# Rif1 inhibits replication fork progression and controls DNA copy number in Drosophila

Alexander Munden[1], Zhan Rong[1], Amanda Sun[1], Rama Gangula[2], Simon Mallal[2,3], Jared T Nordman[1]*

[1]Department of Biological Sciences, Vanderbilt University, Nashville, United States; [2]Department of Medicine, Vanderbilt University School of Medicine, Nashville, United States; [3]Department of Pathology, Microbiology and Immunology, Vanderbilt University School of Medicine, Nashville, United States

**Abstract** Control of DNA copy number is essential to maintain genome stability and ensure proper cell and tissue function. In *Drosophila* polyploid cells, the SNF2-domain-containing SUUR protein inhibits replication fork progression within specific regions of the genome to promote DNA underreplication. While dissecting the function of SUUR's SNF2 domain, we identified an interaction between SUUR and Rif1. Rif1 has many roles in DNA metabolism and regulates the replication timing program. We demonstrate that repression of DNA replication is dependent on Rif1. Rif1 localizes to active replication forks in a partially SUUR-dependent manner and directly regulates replication fork progression. Importantly, SUUR associates with replication forks in the absence of Rif1, indicating that Rif1 acts downstream of SUUR to inhibit fork progression. Our findings uncover an unrecognized function of the Rif1 protein as a regulator of replication fork progression.

DOI: https://doi.org/10.7554/eLife.39140.001

*For correspondence:
jared.nordman@vanderbilt.edu

**Competing interests:** The authors declare that no competing interests exist.

## Introduction

Accurate duplication of a cell's genetic information is essential to maintain genome stability. Proper regulation of DNA replication is necessary to prevent mutations and other chromosome aberrations that are associated with cancer and developmental abnormalities (*Jackson et al., 2014*). DNA replication begins at thousands of cis-acting sites termed origins of replication. The Origin Recognition Complex (ORC) binds to replication origins where, together with Cdt1 and Cdc6, it loads an inactive form of the MCM2-7 replicative helicase (*Bell and Labib, 2016*). Inactive helicases are phosphorylated by two key kinases, S-CDK and Dbf4-dependent kinase (DDK), which results in the activation of the helicase and recruitment of additional factors to form a pair of bi-directional replication forks emanating outward from the origin of replication (*Siddiqui et al., 2013*). Although many layers of regulation control the initiation of DNA replication, much less is known about how replication fork progression is regulated.

In metazoans, replication origins are not sequence specific and are likely specified by a combination of epigenetic and structural features (*Aggarwal and Calvi, 2004*; *Cayrou et al., 2011*; *Eaton et al., 2011*; *Mesner et al., 2011*; *Miotto et al., 2016*; *Remus et al., 2004*). Furthermore, replication origins are not uniformly distributed throughout the genome. The result of non-uniform origin distribution is that, in origin-poor regions of the genome, a single replication fork must travel great distances to complete replication. If a replication fork encounters an impediment within a large origin-less region of the genome, then replication will be incomplete, resulting in genome instability (*Newman et al., 2013*). In fact, origin-poor regions of the genome are known to be associated with

chromosome fragility and genome instability (*Debatisse et al., 2012*; *Durkin and Glover, 2007*; *Letessier et al., 2011*; *Norio et al., 2005*). This highlights the need to regulate both the initiation and elongation phases of DNA replication to maintain genome stability.

DNA replication is also regulated in a temporal manner where specific DNA sequences replicate at precise times during S phase, a process known as the DNA replication timing program. While euchromatin replicates in the early part of S phase, heterochromatin and other repressive chromatin types replicate in the latter portion of S phase (*Gilbert, 2002*; *Rhind and Gilbert, 2013*). Although the process of replication timing has been appreciated for many years, the underlying molecular mechanisms controlling timing have remained elusive. The discovery of factors that regulate the DNA replication timing program, however, demonstrate that replication timing is an actively regulated process.

One factor that regulates replication timing from yeast to humans is Rif1 (Rap1-interacting factor 1). Rif1 was initially identified as a regulator of telomere length in budding yeast (*Hardy et al., 1992*), but this function of Rif1 appears to be specific to yeast (*Xu et al., 2004*). Subsequently, Rif1 has been shown to regulate multiple aspects of DNA replication and repair. In mammalian cells, Rif1 has been shown to regulate DNA repair pathway choice by preventing resection of double-strand breaks and favoring non-homologous end joining (NHEJ) over homologous recombination (*Chapman et al., 2013*; *Di Virgilio et al., 2013*; *Zimmermann et al., 2013*). Rif1 from multiple organisms contains a Protein Phosphatase 1 (PP1) interaction motif and Rif1 is able to recruit PP1 to DDK-activated helicases to inactive them and prevent initiation of replication (*Davé et al., 2014*; *Hiraga et al., 2014*; *Hiraga et al., 2017*).

In yeasts, flies and mammalian cells, Rif1 has been shown to regulate the replication timing program (*Cornacchia et al., 2012*; *Hayano et al., 2012*; *Peace et al., 2014*; *Sreesankar et al., 2015*; *Yamazaki et al., 2012*). The precise mechanism(s) through which Rif1 functions to control replication timing are not fully understood. For example, Rif1 has been show to interact with Lamin and is thought to tether specific regions of the genome to the nuclear periphery (*Foti et al., 2016*). How this activity is related to Rif1's ability to inactivate helicases together with PP1 in controlling the timing program remains obscure.

Studying DNA replication in the context of development provides a powerful method to understand how DNA replication is regulated both spatially and temporally. Although DNA replication is a highly ordered process, it must be flexible enough to accommodate the changes in S phase length and cell cycle parameters that occur as cells differentiate (*Matson et al., 2017*). For example, during *Drosophila* development the length of S phase can vary from ~8 hr in a differentiated mitotic cell to 3 – 4 min during early embryonic cell cycles (*Blumenthal et al., 1974*; *Spradling and Orr-Weaver, 1987*). Additionally, many tissues and cell types in *Drosophila* are polyploid, having multiple copies of the genome in a single cell (*Edgar and Orr-Weaver, 2001*; *Lilly and Duronio, 2005*; *Zielke et al., 2013*).

In polyploid cells, copy number is not always uniform throughout the genome (*Rudkin, 1969*; *Hua and Orr-Weaver, 2017*; *Spradling and Orr-Weaver, 1987*). Both heterochromatin and several euchromatic regions of the genome have reduced DNA copy number relative to overall ploidy (*Nordman et al., 2011*). Underreplicated euchromatic regions of the genome share key features with common fragile sites in that they are devoid of replication origins, late replicating, display DNA damage and are tissue-specific (*Andreyeva et al., 2008*; *Nordman et al., 2014*; *Sher et al., 2012*; *Yarosh and Spradling, 2014*). The presence of underreplication is conserved in mammalian cells, but the mechanism(s) mammalian cells use to promote underreplication is unknown (*Hannibal et al., 2014*). In *Drosophila*, underreplication is an active process that is largely dependent on the distribution of ORC and on the Suppressor of Underreplication protein, SUUR (*Hua et al., 2018*; *Makunin et al., 2002*; *Nordman and Orr-Weaver, 2015*).

Understanding how the SUUR protein functions will significantly increase our understanding of the developmental control of DNA replication. The SUUR protein has a recognizable SNF2-like chromatin remodeling domain at its N-terminus, but based on sequence analysis, this domain is predicted to be defective for ATP binding and hydrolysis (*Makunin et al., 2002*; *Nordman and Orr-Weaver, 2015*). Outside of the SNF2 domain, SUUR has no recognizable motifs or domains, which has hampered a mechanistic understanding of how SUUR promotes underreplication. Recently, however, SUUR was shown to control copy number by directly reducing replication fork progression (*Nordman et al., 2014*). SUUR associates with active replication forks and while loss of SUUR

function results in increased replication fork progression, overexpression of SUUR drastically inhibits fork progression without affecting origin firing (*Nordman et al., 2014*; *Sher et al., 2012*). These findings, together with previous work showing that loss of SUUR function has no influence on ORC binding (*Sher et al., 2012*) and that SUUR associates with euchromatin in an S phase-dependent manner (*Kolesnikova et al., 2013*), further supports SUUR as a direct inhibitor of replication fork progression within specific regions of the genome. The mechanism through which SUUR is recruited to replication forks and how it inhibits their progression remains poorly understood.

Here we investigate how SUUR is recruited to replication forks and how it inhibits fork progression. We show that localization of SUUR to replication forks, but not heterochromatin, is dependent on its SNF2 domain. We identify an interaction between SUUR and the conserved replication factor Rif1, indicating they are in the same protein complex. Importantly, we demonstrate that underreplication is dependent on *Rif1*. Critically, we have shown that Rif1 localizes to replication forks in an SUUR-dependent manner, where it acts downstream of SUUR to control replication fork progression. Our findings provide mechanistic insight into the process of underreplication and define a new function for Rif1 in replication control.

## Results

### The SNF2 domain is essential for SUUR function and replication fork localization

As a first step in understanding the mechanism of SUUR function, we wanted to define how it is localized to replication forks. SUUR has only one conserved domain: a SNF2-like domain in its N-terminal region that is predicted to be defective for ATP binding and hydrolysis (*Makunin et al., 2002*; *Nordman and Orr-Weaver, 2015*). To study the function of SUUR's SNF2 domain, we generated a mutant in which the SNF2 domain was deleted and the resulting mutant protein was expressed under the control of the endogenous *SuUR* promoter. This mutant, *SuUR*$^{\Delta SNF}$, was then crossed to an *SuUR* null mutant so that it was the only form of the the SUUR protein present (*Figure 1A*). We tested the function of the SUUR$^{\Delta SNF}$ mutant protein by assessing its ability to promote underreplication in the larval salivary gland. We purified genomic DNA from larval salivary glands isolated from wandering third instar larvae and generated genome-wide copy number profiles using Illumina-based sequencing. We compared the results we obtained from the *SuUR*$^{\Delta SNF}$ mutant to copy number profiles from wild-type (WT) and *SuUR* null mutant salivary glands. To identify underreplicated domains, we used CNVnator, which identifies copy number variants (CNVs) based on a statistical analysis of read depth (*Abyzov et al., 2011*). To be called as underreplicated, regions must not be called as underreplicated in 0 – 2 hr embryo samples that have uniform copy number and must be larger than 10 kb.

The effect of deleting the SNF2 domain was qualitatively and quantitatively similar to the *SuUR* null mutant. Qualitatively, underreplication was suppressed in the *SuUR*$^{\Delta SNF}$ mutant and the copy number profile was similar to the *SuUR* null mutant (*Figure 1B* and *Figure 1—figure supplement 1*). Quantitatively, out of the 90 underreplicated sites identified in WT salivary glands, 59 were not detected in the *SuUR*$^{\Delta SNF}$ mutant (*Supplementary file 1*) and copy number was significantly increased in the euchromatic underreplicated domains similar to the *SuUR* null mutant (*Figure 1C*). We validated our deep-sequencing findings using quantitative droplet digital PCR (ddPCR) at four underreplicated domains (*Figure 1D*). Our findings show that the SNF2-like domain of SUUR is necessary to promote underreplication.

To determine if the SUUR$^{\Delta SNF}$ protein was still able to associate with chromatin, we localized SUUR and the SUUR$^{\Delta SNF}$ mutant proteins in ovarian follicle cells. During follicle cell development, these cells undergo programmed changes in their cell cycle and DNA replication programs (*Claycomb and Orr-Weaver, 2005*; *Hua and Orr-Weaver, 2017*). At a precise time in their differentiation program, follicle cells cease genomic replication and amplify six defined sites of their genome through a re-replication-based mechanism. Early in this gene amplification process, both initiation and elongation phases of replication are coupled. Later in the process, however, initiation no longer occurs and active replication forks can be visualized by pulsing amplifying follicle cells with 5-ethynyl-2′-deoxyuridine (EdU) (*Claycomb et al., 2002*). Active replication forks resolve into a double-bar structure, where each bar represents a series of active replication forks travelling away from the

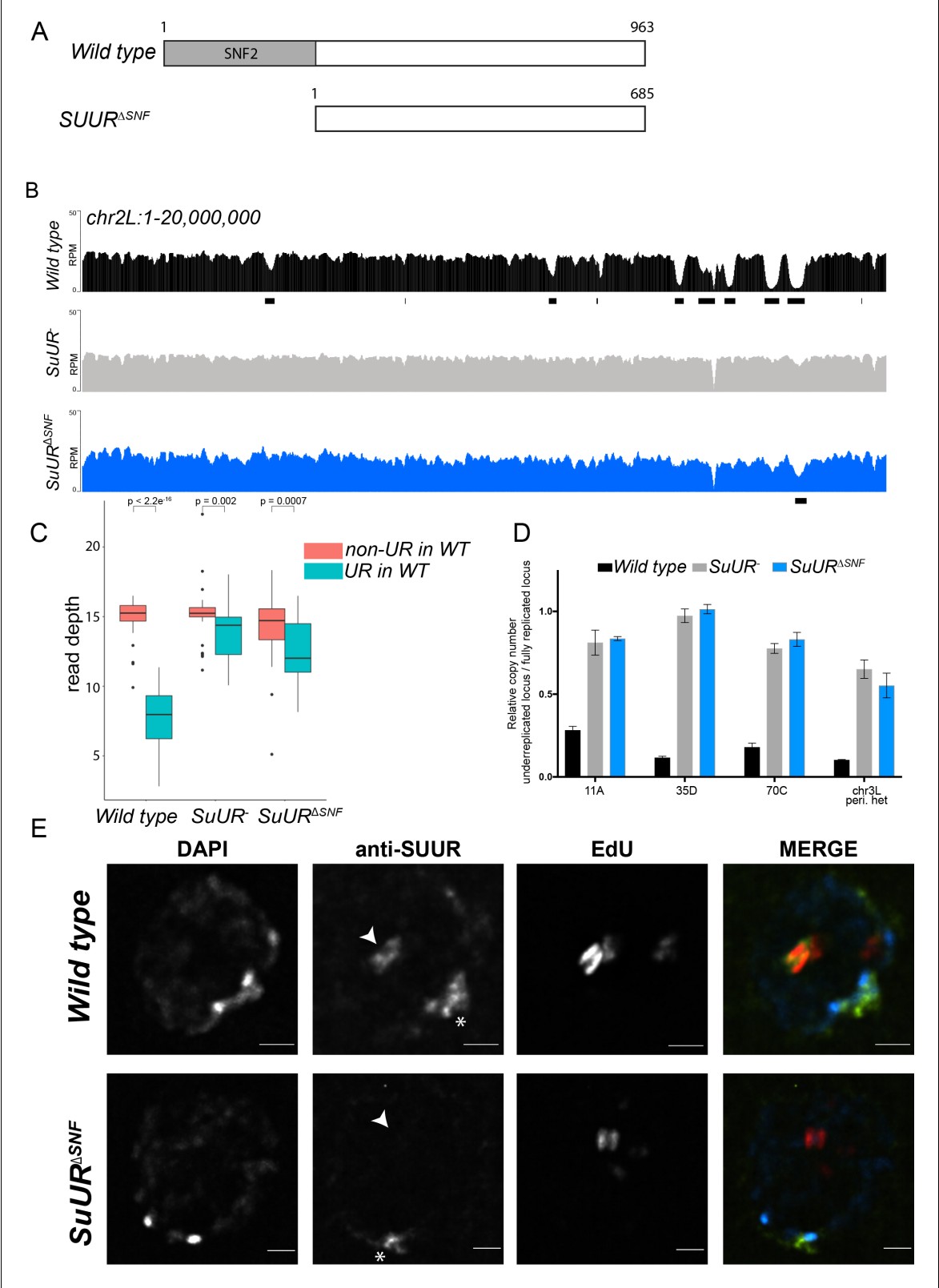

**Figure 1.** The SNF2 domain is essential for SUUR function and replication fork localization. (A) Schematic representation of the SUUR and SUUR$^{\Delta SNF}$ proteins. (B) Illumina-based copy number profiles (Reads Per Million; RPM) of *chr2L* 1 – 20,000,000 from larval salivary glands. Black bars below each profile represent underreplicated regions identified by CNVnator. (C) Average read depth in regions of euchromatic underreplication domains called in wild-type salivary glands vs. the fully replicated regions of the genome. A Welch Two Sample t-test was used to determine p values. (D) Quantitative

*Figure 1 continued on next page*

*Figure 1 continued*

droplet-digital PCR (ddPCR) copy number assay for multiple underreplicated regions. Each bar is the average enrichment relative to a fully replicated control region for three biological replicates. Error bars are the SEM. (E) Localization of SUUR in wild-type and *SuUR*^ΔSNF^ mutant follicle cells. A single representative stage 13 follicle cell nucleus is shown. Arrowheads indicate sites of amplification. Asterisk marks the chromocenter (heterochromatin). Scale bars are 2 μm. DAPI = blue, SUUR = green, EdU = red. The following source data, figure supplements and supplementary files are available for *Figure 1*: *Figure 1—figure supplement 1*; *Figure 1—figure supplement 2*; *Supplementary file 1* - Underreplicated regions called by CNVnator; *Figure 1—source data 1* – Raw data for 1D; *Figure 1—figure supplements 2—source data 1* image intensity data.

DOI: https://doi.org/10.7554/eLife.39140.002

The following source data and figure supplements are available for figure 1:

**Source data 1.** ddPCR data for *Figure 1D*.
DOI: https://doi.org/10.7554/eLife.39140.006
**Figure supplement 1.** Genome-wide copy number profile of the *SuUR*^ΔSNF^ mutant.
DOI: https://doi.org/10.7554/eLife.39140.003
**Figure supplement 2.** Quantification of SUUR and *SUUR*^ΔSNF^ signal intensities at replication forks and heterochromatin.
DOI: https://doi.org/10.7554/eLife.39140.004
**Figure supplement 2—source data 1.** SUUR signal intensity at double bar structures and heterochromatin - raw data.
DOI: https://doi.org/10.7554/eLife.39140.005

origin of replication (*Claycomb and Orr-Weaver, 2005*). By monitoring SUUR localization in amplifying follicle cells, we can unambiguously determine if SUUR associates with active replication forks.

SUUR has two distinct modes of chromatin association during the endo cycle. It constitutively localizes to heterochromatin and dynamically associates with replication forks (*Kolesnikova et al., 2013*; *Nordman et al., 2014*; *Swenson et al., 2016*). In agreement with previous studies, SUUR localized to both replication forks and heterochromatin in amplifying follicle cells (*Figure 1E*) (*Nordman et al., 2014*). In contrast, the SUUR^ΔSNF^ mutant localized to heterochromatin, but its recruitment to active replication forks was severely reduced (*Figure 1E*; *Figure 1—figure supplement 2*). Together, these results demonstrate that the SNF2 domain is important for SUUR recruitment to replication forks and is essential for SUUR-mediated underreplication.

## SUUR associates with Rif1

Interestingly, overexpression of the SNF2 domain and C-terminal portion of SUUR have different underreplication phenotypes. Whereas overexpression of the C-terminal two-thirds of SUUR promotes underreplication (*Kolesnikova et al., 2005*), overexpression of the SNF2 domain suppresses underreplication in the presence of endogenous SUUR (*Kolesnikova et al., 2005*). The C-terminal region of SUUR, however, has no detectable homology or conserved domains (*Makunin et al., 2002*). These observations, together with our own results demonstrating that the SNF2 domain of SUUR is responsible for its localization to replication forks, led us to hypothesize that SUUR is recruited to replication forks through its SNF2 domain where it could recruit an additional factor(s) through its C-terminus to inhibit replication fork progression.

To test the hypothesis that a critical factor interacts with the C-terminal region of SUUR to promote underreplication, we used immunoprecipitation mass spectrometry studies to identify SUUR-interacting proteins. We generated flies that expressed FLAG-tagged full-length SUUR or the SNF2 domain of SUUR under control of the *hsp70* promoter, induced and immunoprecipitated these constructs and identified associated proteins through mass spectrometry. We verified that both full-length SUUR and the SNF2 domain were expressed equally (*Figure 2—figure supplement 1B*). If SUUR recruits a factor to replication forks outside of its SNF2 domain, then we would expect this factor to be present only in full-length purifications and not in the SNF2 domain purification. A single protein fulfilled this criteria: Rif1 (*Figure 2A*). This result raises the possibility Rif1 works together with SUUR to inhibit replication fork progression.

To ensure that the association between SUUR and Rif1 was not bridged by chromatin, we used NP40 to extract chromatin proteins and treated the extract with Benzonase to digest DNA. We then immunoprecipitated FLAG-SUUR and used Western blotting to determine if Rif1 could co-IP using a highly specific anti-Rif1 antibody (*Figure 2—figure supplement 2*). Even in these conditions, SUUR was able to co-IP Rif1 (*Figure 2B*; *Figure 2—figure supplement 1A*). We conclude that SUUR and

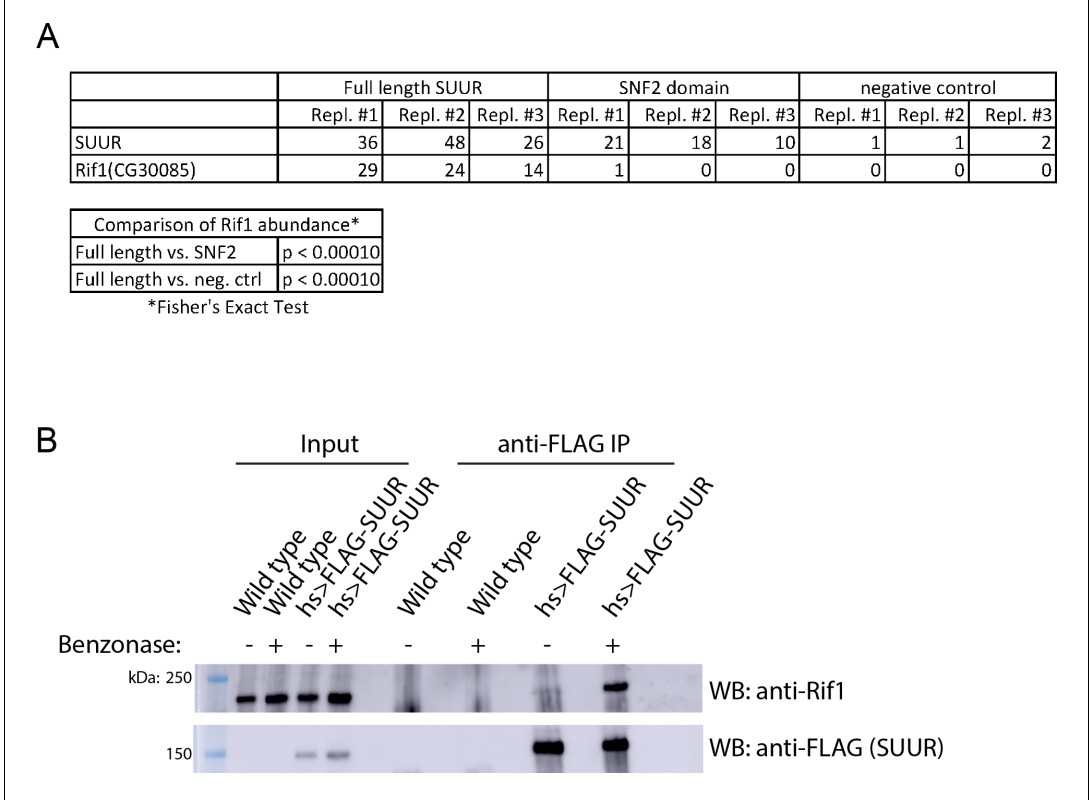

**Figure 2.** SUUR associates with Rif1. (**A**) Total spectrum counts of FLAG-SUUR, FLAG-SNF2 and Oregon R (no FLAG control) for three independent IP-mass spectrometry experiments (biological replicates). A Fisher's Exact test of spectrum counts was used to determine significance. (**B**) Immunoprecipitation of FLAG-SUUR and no FLAG control (wild-type) from 0 to 24 hr embryos extracted with NP40 lysis buffer with or without Benzonase treatment. Membranes were probed with anti-Rif1 and anti-FLAG antibodies to monitor Rif1 and SUUR, respectively. The following source data and figure supplement are available for *Figure 2*: *Figure 2—figure supplement 1*; *Figure 2—figure supplement 2*; *Figure 2—source data 1* – SUUR mass spectrometry total spectrum counts; *Figure 2—figure supplements 2—source data 1* – embryo hatch rata data.
DOI: https://doi.org/10.7554/eLife.39140.007

The following source data and figure supplements are available for figure 2:

**Source data 1.** Results of SUUR IP-mass spec screen.
DOI: https://doi.org/10.7554/eLife.39140.011

**Figure supplement 1.** Western blot analysis of heat-shock inducible SUUR constructs.
DOI: https://doi.org/10.7554/eLife.39140.008

**Figure supplement 2.** Verification of *Rif1* mutants and validation of anti-Rif1 antibody.
DOI: https://doi.org/10.7554/eLife.39140.009

**Figure supplement 2—source data 1.** Raw data for hatch rate assay.
DOI: https://doi.org/10.7554/eLife.39140.010

Rif1 exist in the same protein complex and the interaction between SUUR and Rif1 is independent of chromatin bridging.

## Underreplication is dependent on Rif1

If SUUR recruits Rif1 to replication forks to promote underreplication, then underreplication should be dependent on *Rif1*. To test this hypothesis, we used CRISPR-based mutagenesis to generate *Rif1* null mutants in *Drosophila* (*Bassett et al., 2013*; *Gratz et al., 2013*) (*Figure 3A*). Western blot analysis of ovary extracts from two deletion mutants, $Rif1^1$ and $Rif1^2$, show no detectable Rif1 protein (*Figure 2—figure supplement 2A*). Also, no signal was detected in the $Rif1^1/Rif1^2$ mutant by immunofluorescence (*Figure 2—figure supplement 2B*). The $Rif1^1/Rif1^2$ null mutant was viable and fertile showing only a modest defect in embryonic hatch rate relative to wild-type flies with a 92% hatch rate for wild-type embryos vs. 88% for the $Rif1^1/Rif^2$ mutant embryos (*Figure 2—figure supplement*

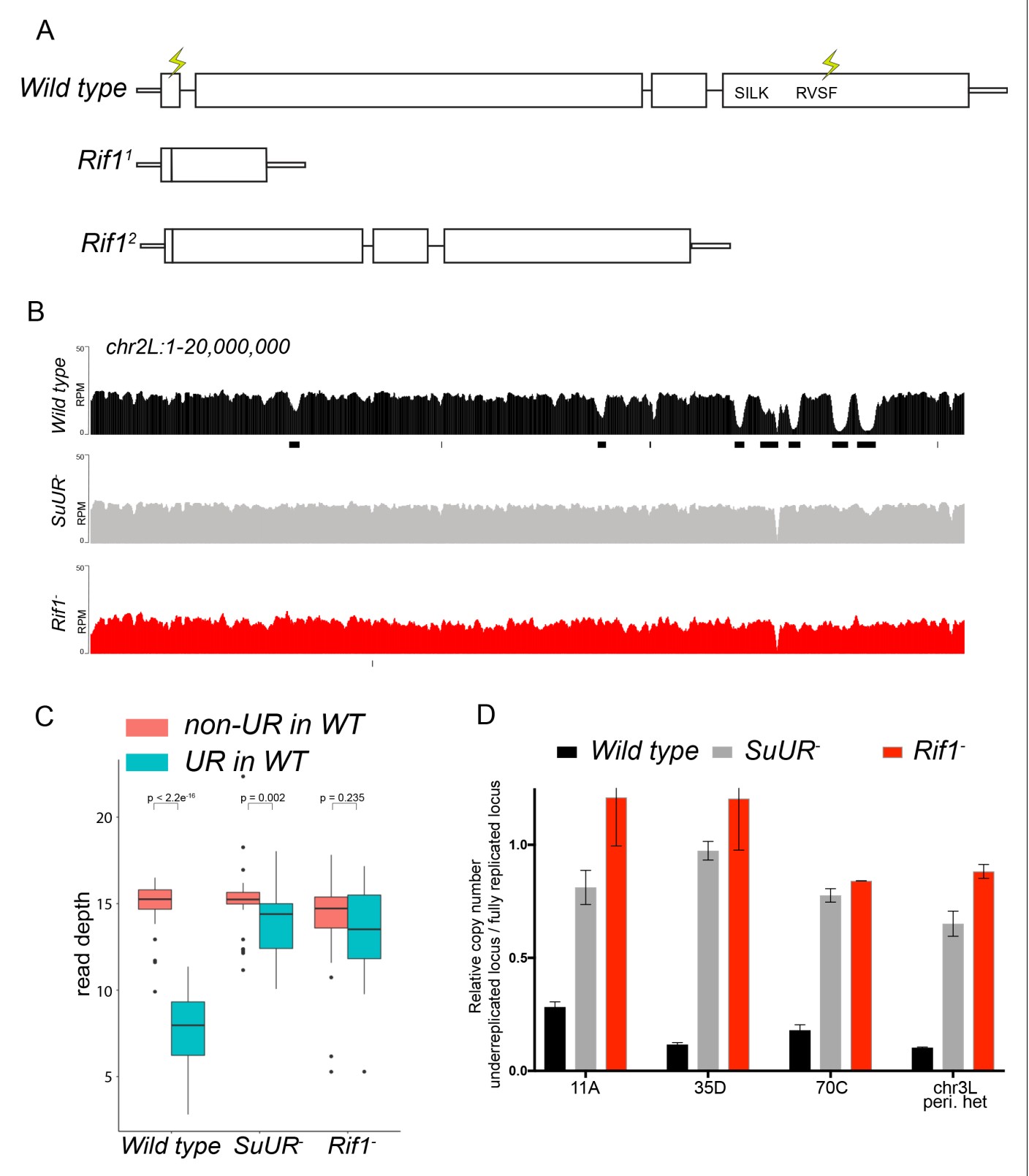

**Figure 3.** Rif1 is required for underreplication. (**A**) Schematic representation of the *Rif1* gene and CRISPR-induced *Rif1* mutants. Lightning bolts represent the 5' and 3' gRNA positions. (**B**) Illumina-based copy number profiles of the *chr2L* from larval salivary glands. Black bars below each profile represent underreplicated regions identified by CNVnator. The wild-type and *SuUR* profiles are the same as in *Figure 1b*. (**C**) Average read depth in regions of euchromatic underreplication domains called in wild-type salivary glands vs. the fully replicated regions of the genome. A Welch Two

*Figure 3 continued on next page*

*Figure 3 continued*

Sample t-test was used to determine p values. (**D**) Quantitative droplet-digital PCR (ddPCR) copy number assay for multiple underreplicated regions. Each bar is the average enrichment relative to a fully replicated control region for three biological replicates. Error bars are the SEM. The following source data and figure supplement are available for *Figure 3*: *Figure 3—figure supplement 1*; *Figure 3—figure supplement 2*; *Figure 3—figure supplement 2*; *Figure 3—source data 1*; *Figure 3—figure supplement 2—source data 1* - Raw data for 2B.
DOI: https://doi.org/10.7554/eLife.39140.012

The following source data and figure supplements are available for figure 3:

**Source data 1.** ddPCR data for *Figure 3D*.
DOI: https://doi.org/10.7554/eLife.39140.017
**Figure supplement 1.** Genome-wide copy number profile of the *Rif1* mutant.
DOI: https://doi.org/10.7554/eLife.39140.013
**Figure supplement 2.** Rif1 mutant salivary gland cells display a pattern of late replication.
DOI: https://doi.org/10.7554/eLife.39140.014
**Figure supplement 2—source data 1.** Raw data for *Figure 3—figure supplement 2B*.
DOI: https://doi.org/10.7554/eLife.39140.015
**Figure supplement 3.** *Rif1* mutant endo cycling cells have enlarged chromocenters.
DOI: https://doi.org/10.7554/eLife.39140.016

*2C*). This is in contrast to a previous study reporting *Rif1* is essential in *Drosophila* (*Sreesankar et al., 2015*) and consistent with a recent study that generated an independent *Rif1* null mutant using CRISPR-based mutagenesis (*Seller and O'Farrell, 2018*). Rif1's essentiality, however, was based on RNAi and not a mutation of the *Rif1* gene (*Sreesankar et al., 2015*). The most likely explanation for this discrepancy is that the lethality in the RNAi experiments was due to an off-target effect.

To determine if *Rif1* is necessary for underreplication, we dissected salivary glands from *Rif1¹/Rif1²* (herein referred to as *Rif1⁻*) heterozygous larvae and extracted genomic DNA for Illumina-based sequencing to measure changes in DNA copy number. Strikingly, underreplication is abolished upon loss of Rif1 function (*Figure 3B and C*; *Figure 3—figure supplement 1*). We validated our sequence-based copy number assays with quantitative PCR at a subset of underreplicated regions using ddPCR (*Figure 3D*). Furthermore, we determined the read density at all euchromatic sites of underreplication called in our wild-type samples, which quantitatively demonstrates that Rif1 is essential for underreplication (*Figure 3C*). These results demonstrate that underreplication is dependent on *Rif1*.

It is possible that the *Rif1* mutant indirectly influences underreplication through changes in replication timing. Underreplicated domains, both euchromatic and heterochromatic, tend to be late replicating regions of the genome (*Belyaeva et al., 2012*; *Makunin et al., 2002*). Therefore, if these regions replicated earlier in S phase in a *Rif1* mutant, then this change could prevent their underreplication. In fact, SUUR associates with late replicating regions of the genome (*Filion et al., 2010*; *Pindyurin et al., 2007*). Due to their large polyploid nature, salivary gland cells cannot be sorted to perform genome-wide replication timing experiments. Because heterochromatin replicates exclusively in late S phase, however, late replication can be visualized when EdU is incorporated exclusively in regions of heterochromatin. To assess if *Rif1* mutants have a clear pattern of late replication in larval salivary glands, we isolated salivary glands from early 3rd instar larvae, which are actively undergoing endo cycles. We pulsed these salivary glands with EdU to visualize sites of replication and co-stained with an anti-HP1 antibody to mark heterochromatin. In wild-type salivary glands, only rarely (1 of 238 EdU⁺ cells; 0.4%) did we detect EdU incorporation in regions of heterochromatin (*Figure 3—figure supplement 2*). This is consistent with the lack of heterochromatin replication due to underreplication. In contrast, in both *SuUR* and *Rif1* mutants, we could readily detect cells that were solely incorporating EdU within regions of heterochromatin (32 of 327 EdU⁺ cells; 9.8% for *SuUR* and 70 of 385 EdU⁺ cells; 18.2% for *Rif1*) (*Figure 3—figure supplement 2*). Therefore, we conclude that *Rif1* mutants still have a clear pattern of late replication. Given that heterochromatin underreplication is suppressed in a *Rif1* mutant, although it is still late replicating, indicates that replication timing cannot solely explain the lack of underreplication associated with loss of Rif1 function.

While characterizing Rif1's role in underreplication and patterns of DNA replication in endo cycling cells, we did observe differences in the heterochromatic regions of *SuUR* and *Rif1* mutants.

First, although underreplication is suppressed in both mutants (*Figure 3* and *Figure 3—figure supplement 1*), the chromocenters were abnormally large in the *Rif1* mutant relative to an *SuUR* mutant as observed by DAPI staining consistent with the 'fluffy' enlarged chromocenters seen in Rif1 mutant mouse cells (*Figure 3—figure supplement 3*) (*Cornacchia et al., 2012*). Although, this phenotype was present in all endo cycling cells, it was especially dramatic in the ovarian nurse cells (*Figure 3—figure supplement 3*). Second, Illumina-based copy number profiles revealed an increase in copy number in some pericentric heterochromatin regions in the *Rif1* mutant relative to the *SuUR* mutant (*Figure 3—figure supplement 1*). Collectively, these results suggest that heterochromatin is partially, but not fully replicated in *SuUR* mutant endo cycling cells, consistent with previous cytological analysis (*Demakova et al., 2007*). In contrast, loss of Rif1 function appears to completely restore heterochromatic replication in endo cycling cells.

## Rif1 affects replication fork progression

SUUR-mediated underreplication occurs through inhibition of replication fork progression (*Nordman et al., 2014*; *Sher et al., 2012*). If SUUR acts together with Rif1 to promote underreplication, then Rif1 is expected to control replication fork progression. DNA combing assays in human and mouse cells from multiple groups have come to different conclusions as to whether Rif1 affects replication fork progression (*Alver et al., 2017*; *Cornacchia et al., 2012*; *Hiraga et al., 2017*; *Yamazaki et al., 2012*). Rif1, however, has been shown to be associated with replication forks through nascent chromatin capture, an iPOND-like technique used to isolate proteins associated with active replication forks (*Alabert et al., 2014*). To determine directly if Rif1 controls replication fork progression, we performed copy number assays on amplifying follicle cells.

Gene amplification in ovarian follicle cells occurs at six discrete sites in the genome through a re-replication based mechanism. Copy number profiling of these amplified domains provides a quantitative assessment of the number of rounds of origin firing and the distance replication forks have travelled during the amplification process, allowing us to disentangle the initiation and elongation phases of DNA replication. To determine if Rif1 affects origin firing and/or replication fork progression, we isolated wild-type and *Rif1* mutant stage 13 egg chambers, which represent the end point of the amplification process, and made quantitative DNA copy number measurements. Loss of Rif1 function resulted in an increase in replication fork progression without significantly affecting copy number at the origin of replication at all sites of amplification (*Figure 4A*). The increase in fork progression observed in the *Rif1* mutant was not due to a lengthening of the developmental time window for gene amplification, as there was no significant difference in egg chamber distribution between wild-type and *Rif1* mutant ovaries (*Figure 4—figure supplement 1*).

To quantify the changes in fork progression we observed at sites of amplification, we computationally determined the peak of amplification and the region on each arm of the amplified domain that represents one half of the copy number at the highest point of the amplicon (*Nordman et al., 2014*). This quantitative analysis of origin firing and replication fork progression revealed that origin firing was not affected in the *Rif1* mutant, as no major change in copy number was detected at the origin of replication when comparing wild type and *Rif1* mutant stage 13 follicle cells (*Supplementary file 2*). In contrast, the width of each replication gradient, which represents the rate of fork progression, was significantly increased at all sites of amplification (*Figure 4A*; *Supplementary file 2*). Based on the observation that the *Rif1* mutant does not affect origin firing, but specifically affects the distance replication forks travel during the gene amplification process, and there was no change in the developmental window of gene amplification in the *Rif1* mutant, we conclude that Rif1 regulates replication fork progression.

Given that the *Rif1* mutant phenocopies an *SuUR* mutant with respect to replication fork progression, we next wanted to determine the cause of increased replication fork progression at amplified loci upon loss of Rif1 function. Previously, it was shown that a prolonged period of EdU incorporation in the *SuUR* mutant, within the 7.5 hr span of gene amplification, gives rise to the extended replication gradient at sites of amplification (*Nordman et al., 2014*). Gene amplification starts synchronously in all follicle cells at stage 10B of egg chamber development (*Calvi et al., 1998*). By the end of gene amplification, however, only a subset of follicle cells display visual amplification foci as judged by EdU incorporation, likely representing a stochastic end to the gene amplification process (*Nordman et al., 2014*). To determine if Rif1 controls replication fork progression by increasing the period of EdU incorporation within the 7.5 hr time window of gene amplification, comparable to an

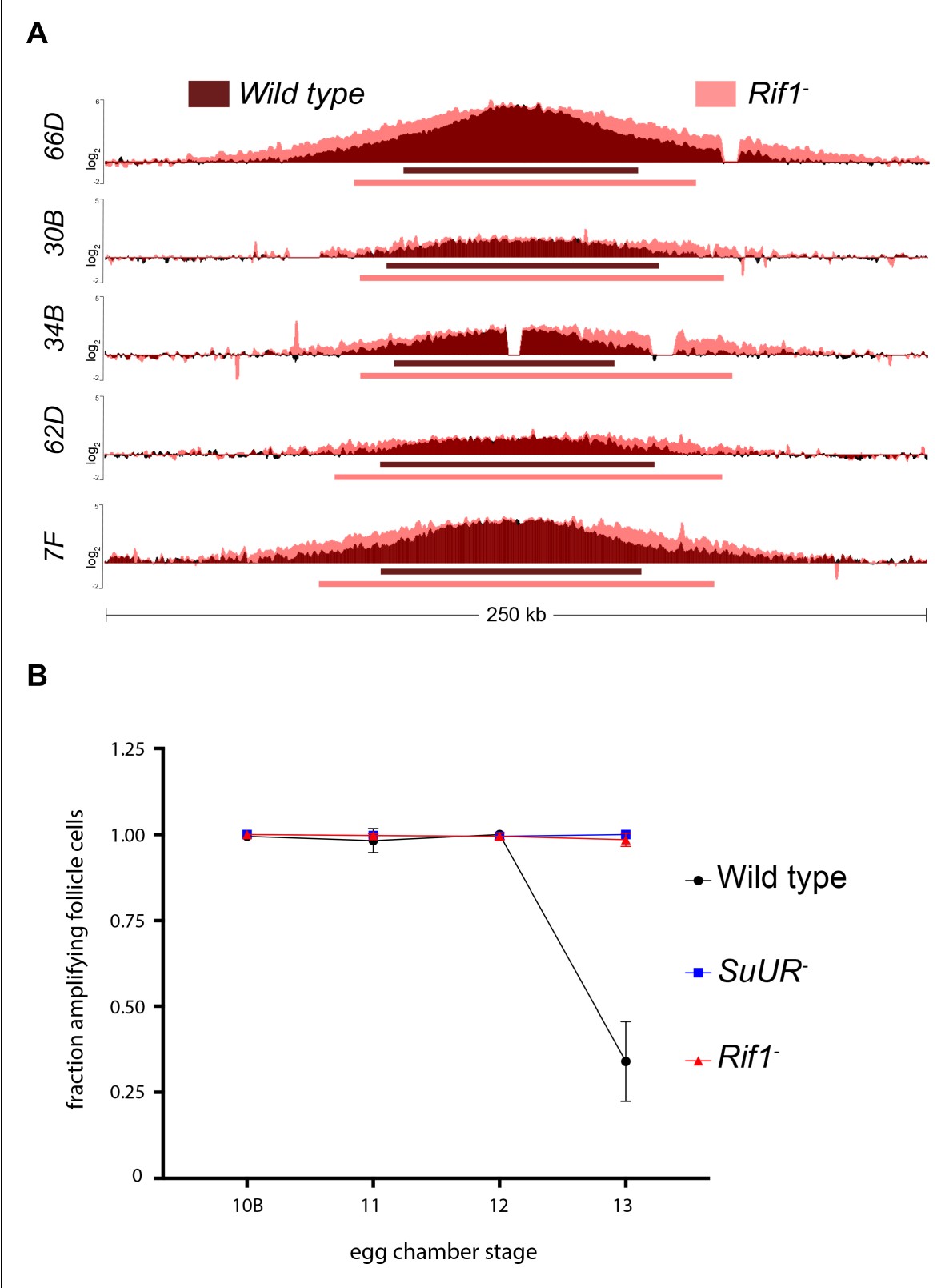

**Figure 4.** Rif1 regulates replication fork progression. (**A**) Illumina-based copy number profile of sites of follicle cell gene amplification. DNA was extracted from wild type and *Rif1* mutant stage 13 egg chambers and compared to DNA extracted from 0 to 2 hr embryos. The resulting graphs are the log$_2$-transformed ratios of egg chamber relative to embryonic DNA. Bars below the graphs represent the distance between the half-maximum copy number on each side of the replication origin. (**B**) Fraction of cells that display visible amplification foci in each stage of gene amplification. Average of

*Figure 4 continued on next page*

*Figure 4 continued*

two biological replicates in which two egg chambers from each stage were used per biological replicate. 100 – 300 follicle cells were counted per genotype. Error bars are the SEM. The following source data, supplementary file and figure supplement are available for *Figure 4*: *Figure 4—figure supplement 1*; *Supplementary file 2* – Table of half-max values for all amplicons; *Figure 4—source data 1* – raw data for 4B; *Figure 4—figure supplements 1—source data 1*-combined data for egg chamber distribution of five biological replicates.
DOI: https://doi.org/10.7554/eLife.39140.018
The following source data and figure supplements are available for figure 4:

**Source data 1.** Data for *Figure 4B*.
DOI: https://doi.org/10.7554/eLife.39140.021
**Figure supplement 1.** The developmental window of gene amplification is not affected by loss of Rif1 function.
DOI: https://doi.org/10.7554/eLife.39140.019
**Figure supplement 1—source data 1.** Raw data for egg chamber distribution assay.
DOI: https://doi.org/10.7554/eLife.39140.020

*SuUR* mutant, we quantified the fraction of stage 13 follicle cells that were EdU positive. Similar to an *SuUR* mutant, loss of Rif1 function also resulted in a prolonged period of EdU incorporation with 34% of follicle cells visibly incorporating EdU in wild-type follicle cells, 100% in an *SuUR* mutant and 98.5% in the *Rif1* mutant (*Figure 4B*). This result suggests that Rif1 has a destabilizing effect on replication forks, resulting in a premature cessation of replication fork progression.

## Rif1 acts downstream of SUUR

Rif1 could control SUUR activity and underreplication by at least two different mechanisms. Rif1 could act upstream of SUUR and directly or indirectly regulate SUUR's ability to associate with chromatin. For example, Histone H1 and HP1 affect underreplication by influencing SUUR's ability to associate with chromatin (*Andreyeva et al., 2017*; *Pindyurin et al., 2008*). Alternatively, Rif1 could act downstream of SUUR to control replication fork progression. We sought to distinguish between these possibilities by determining whether SUUR could still associate with replication forks in the absence of Rif1 function.

To monitor SUUR's association with heterochromatin and replication forks in the same cell type, we localized SUUR in amplifying follicle cells where replication forks (double bars) and heterochromatin (chromocenter) can be visualized unambiguously, in the presence and absence of Rif1. SUUR localized to both replication forks and heterochromatin in the absence of Rif1 function (*Figure 5*; *Figure 5—figure supplement 1*). Therefore, we conclude that Rif1 acts downstream of SUUR to inhibit fork progression and that SUUR lacks the ability to inhibit replication fork progression in the absence of Rif1.

## Rif1 localizes to active replication forks

Although our genetic data indicate that Rif1 affects replication fork progression, we wanted to determine if Rif1 controls replication fork progression through a direct or indirect mechanism. If Rif1 directly influences replication fork progression and/or stability, then it should localize to active replication forks. To assess this possibility, we visualized Rif1 localization during gene amplification in follicle cells using a Rif1-specific antibody (*Figure 2—figure supplement 1*).

Rif1 localization pattern was strikingly similar to that of SUUR. First, Rif1 is localized to heterochromatin in all stages of amplifying follicle cells (*Figure 6*). Second, Rif1 localized to sites of amplification even prior to the formation of double bar structures (*Figure 6*; *Figure 6—figure supplement 1*). Third, in the later stages of gene amplification Rif1 was localized to active replication forks. Taken together, these results demonstrate that Rif1 dynamically associates with replication forks to regulate their progression.

To verify that Rif1 associates with replication forks in a context other than the gene amplification, we used iPOND to determine if Rif1 is associated with replication forks in cultured *Drosophila* S2 cells. Briefly, cells were pulsed with EdU and immediately fixed in formaldehyde or chased with thymidine prior to fixation. Proteins associated with newly synthesized DNA (replication forks) can be identified based on their enrichment in pulse samples relative to chase samples (*Dungrawala and Cortez, 2014*; *Sirbu et al., 2011*). We used mass spectrometry to quantify Rif1 protein abundance in EdU pulse and chase samples (*Sirbu et al., 2013*). Consistent with Rif1 association with replication

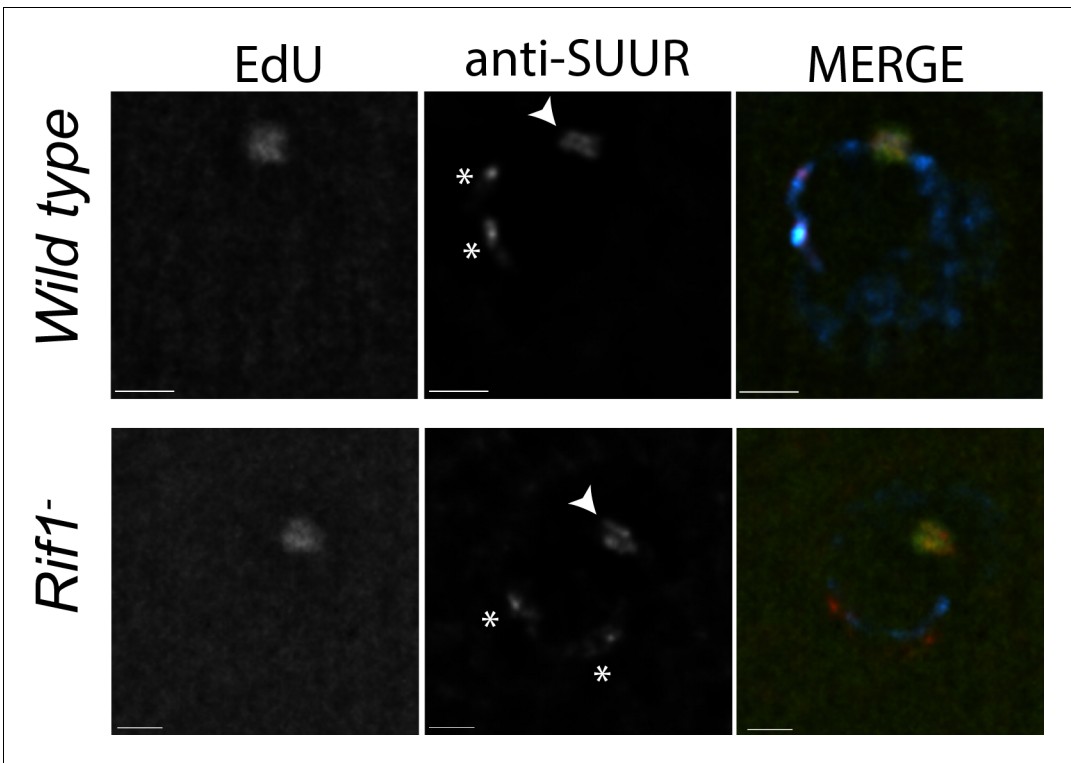

**Figure 5.** Rif1 acts downstream of SUUR. Localization of replication forks (EdU) and SUUR in a wild-type and *Rif1* mutant follicle cell nuclei. A single representative stage 13 follicle cell nucleus is shown. Scale bars are 2 μm. Arrowheads indicate sites of amplification. Asterisks marks the chromocenter (heterochromatin). DAPI = blue, SUUR = green, EdU = red. The following source data and figure supplement are available for *Figure 5*: *Figure 5—figure supplement 1*; *Figure 5—figure supplements 1—source data 1*–intensity data.
DOI: https://doi.org/10.7554/eLife.39140.022

The following source data and figure supplements are available for figure 5:

**Figure supplement 1.** Quantification of SUUR signal intensity at replication forks in the presence and absence of Rif1.
DOI: https://doi.org/10.7554/eLife.39140.023

**Figure supplement 1—source data 1.** SUUR signal intensity at double bar structures - raw data.
DOI: https://doi.org/10.7554/eLife.39140.024

---

forks in amplifying follicle cells, Rif1 was enriched in EdU pulse samples relative to chase samples in cultured cells (*Figure 6—figure supplement 2A*). Although this enrichment was not as abundant as our PCNA-positive control, this is expected for a protein that associates with a subset of replication forks.

To independently verify that Rif1 is localized to replication forks in cultured cells, we performed a proximity ligation assay (PLA)-based approach with nascent DNA (*Roy et al., 2018*; *Taglialatela et al., 2017*). *Drosophila* S2 cells were pulsed with EdU, fixed and EdU was subsequently biotinylated. A PLA assay was then performed using two different anti-biotin antibodies as a positive control, or an anti-biotin antibody together with an anti-Rif1 antibody. As a negative control, the same PLA assays were performed using cells that were not pulsed with EdU. Consistent with our iPOND mass-spec results, PLA foci were generated using anti-Rif1 and anti-biotin antibodies only when cells were pulsed with EdU (*Figure 6—figure supplement 2B*). Together, these results indicate that Rif1 is associated with replication forks in amplifying follicle cells and cultured cells.

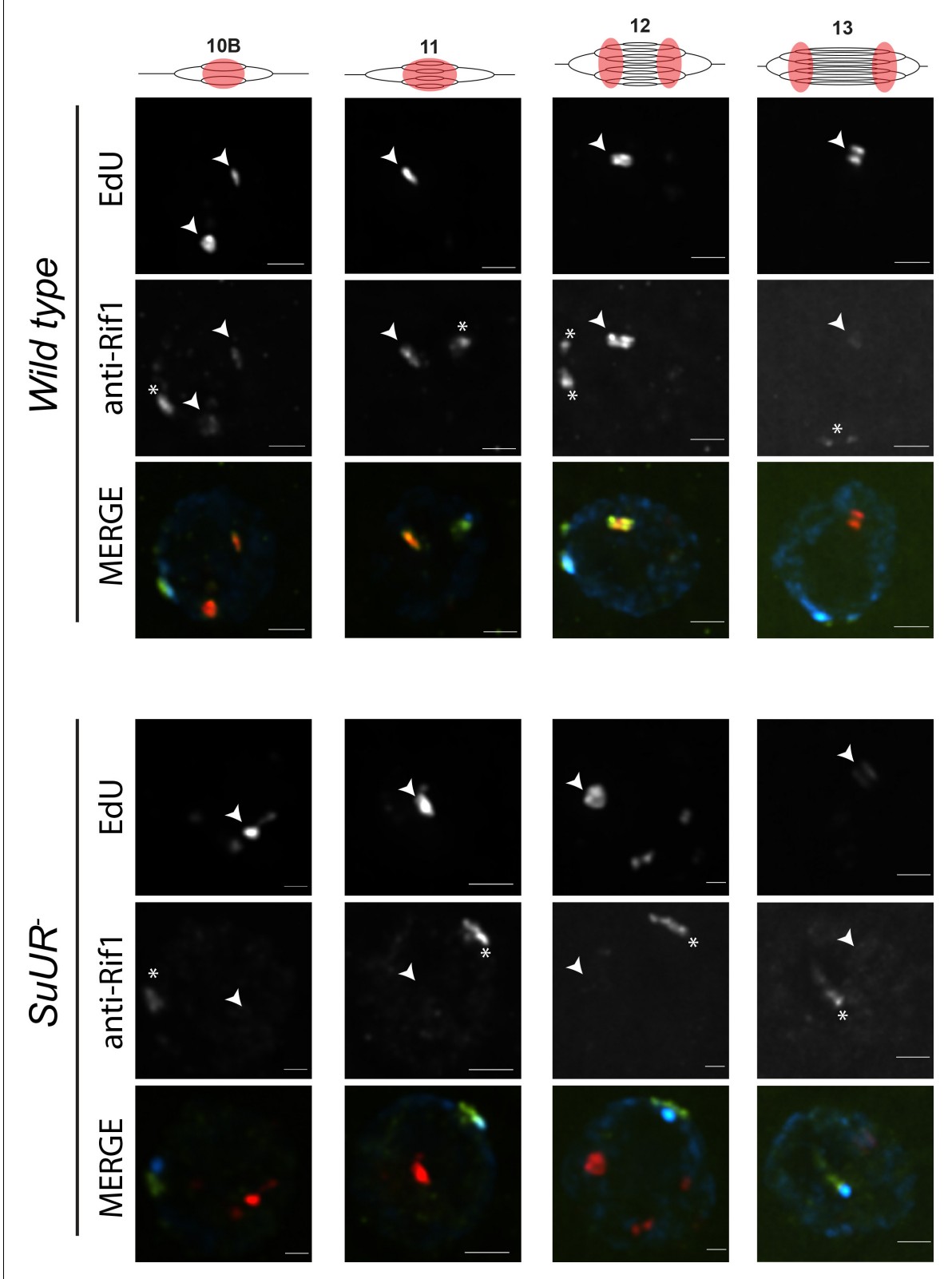

**Figure 6.** SUUR is necessary to retain Rif1 at replication forks. Localization of active replication forks (EdU) and Rif1 in a wild-type and *SuUR* mutant follicle cell nuclei. Single representative follicle cell nuclei are shown for each stage. Scale bars are 2 μm. Arrowheads indicate sites of amplification. Asterisk marks the chromocenter (heterochromatin). The following source data and figure supplement are available for *Figure 6*: *Figure 6—figure*

*Figure 6 continued on next page*

*Figure 6 continued*

*supplement 1*; *Figure 6—figure supplement 2*; *Figure 6—figure supplements 1—source data 1*–intensity data; *Figure 6—figure supplement 2 –* raw data for 2B.
DOI: https://doi.org/10.7554/eLife.39140.025
The following source data and figure supplements are available for figure 6:
**Figure supplement 1.** Quantification of Rif1 signal intensity at replication forks in the presence and absence of SUUR.
DOI: https://doi.org/10.7554/eLife.39140.026
**Figure supplement 1—source data 1.** Rif1 signal intensity at amplificaiton loci - stages 10B-13 - raw data.
DOI: https://doi.org/10.7554/eLife.39140.027
**Figure supplement 2.** Rif1 localizes to replication forks in cultured cells.
DOI: https://doi.org/10.7554/eLife.39140.028
**Figure supplement 2—source data 1.** Data for *Figure 6—figure supplement 2B*.
DOI: https://doi.org/10.7554/eLife.39140.029

## SUUR is required to retain Rif1 at replication forks

Based on our observations that SUUR and Rif1 are part of the same protein complex, and that a *Rif1* mutant phenocopies an *SuUR* mutant, we hypothesized that SUUR recruits a Rif1/PP1 complex to replication forks. If true, then Rif1 association with replication forks should be at least partially dependent on SUUR. To test this hypothesis, we monitored the localization of Rif1 in *SuUR* mutant amplifying follicle cells. We found that Rif1's association with replication forks was largely dependent on SUUR, as the Rif1 signal was lost in late stage amplifying follicle cells in an *SuUR* mutant (*Figure 6*; *Figure 6—figure supplement 1*). Rif1's recruitment to replication foci, however, was not completely dependent on SUUR. In a subset of stage 10B and 11 egg chambers, when both initiation of replication and fork progression are still coupled, we observed Rif1 localization to amplification foci in a subset of follicle cells (*Figure 6—figure supplement 1*). Rif1 staining was lost, however, in stage 12 and 13 egg chambers. We conclude that while the initial recruitment of Rif1 to sites of amplification is not completely dependent on SUUR, SUUR is necessary to retain Rif1 at replication forks.

## The PP1-interacting motif of Rif1 is necessary for underreplication

Because Rif1 is known to recruit PP1 to replication origins to regulate initiation, this led us to ask if the same interaction between Rif1 and PP1 is important for Rif1's regulation of replication fork progression. Rif1 associates with Protein Phosphatase 1 (PP1) through a conserved interaction motif, thereby recruiting PP1 to MCM complexes and inactivating them (*Davé et al., 2014*; *Hiraga et al., 2017*; *Hiraga et al., 2014*). PP1 has also been shown to associate with Rif1 in *Drosophila* (*Seller and O'Farrell, 2018*; *Sreesankar et al., 2015*). Based on this model of Rif1 function, we wanted to determine if Rif1's PP1 interaction motif was necessary for Rif1-mediated underreplication. We used CRISPR-based mutagenesis to mutate the conserved SILK/RSVF PP1 interaction motif to SAAK/RASA. Western blot analysis showed that mutation of the SILK/RSVF motif did not affect protein stability (*Figure 7—figure supplement 1*). Mutation of this motif has been shown to disrupt the Rif1/PP1 interaction in organisms from yeast to humans (*Alver et al., 2017*; *Davé et al., 2014*; *Hiraga et al., 2017*; *Hiraga et al., 2014*; *Mattarocci et al., 2014*; *Sreesankar et al., 2015*; *Sukackaite et al., 2017*). We isolated salivary glands from wandering 3rd instar larvae *of the Rif1^PP1* mutant and *Rif1^PP1*/+ heterozygous animals as a wild-type control. We then extracted DNA and generated genome-wide copy number profiles by Illumina sequencing. Similar to the *Rif1* mutant, underreplication was largely abolished in the *Rif1^PP1* mutant (*Figure 7A–C*; *Supplementary file 1*). Thus, Rif1's PP1-interaction motif is necessary to promote underreplication, suggesting that PP1 is a mediator of underreplication. It still remains possible, however, that an additional protein(s) could interact with this motif to promote underreplication.

## Discussion

The SUUR protein is responsible for promoting underreplication of heterochromatin and many euchromatin regions of the genome. Although SUUR was recently shown to promote underreplication through inhibition of replication fork progression, the underlying molecular mechanism has

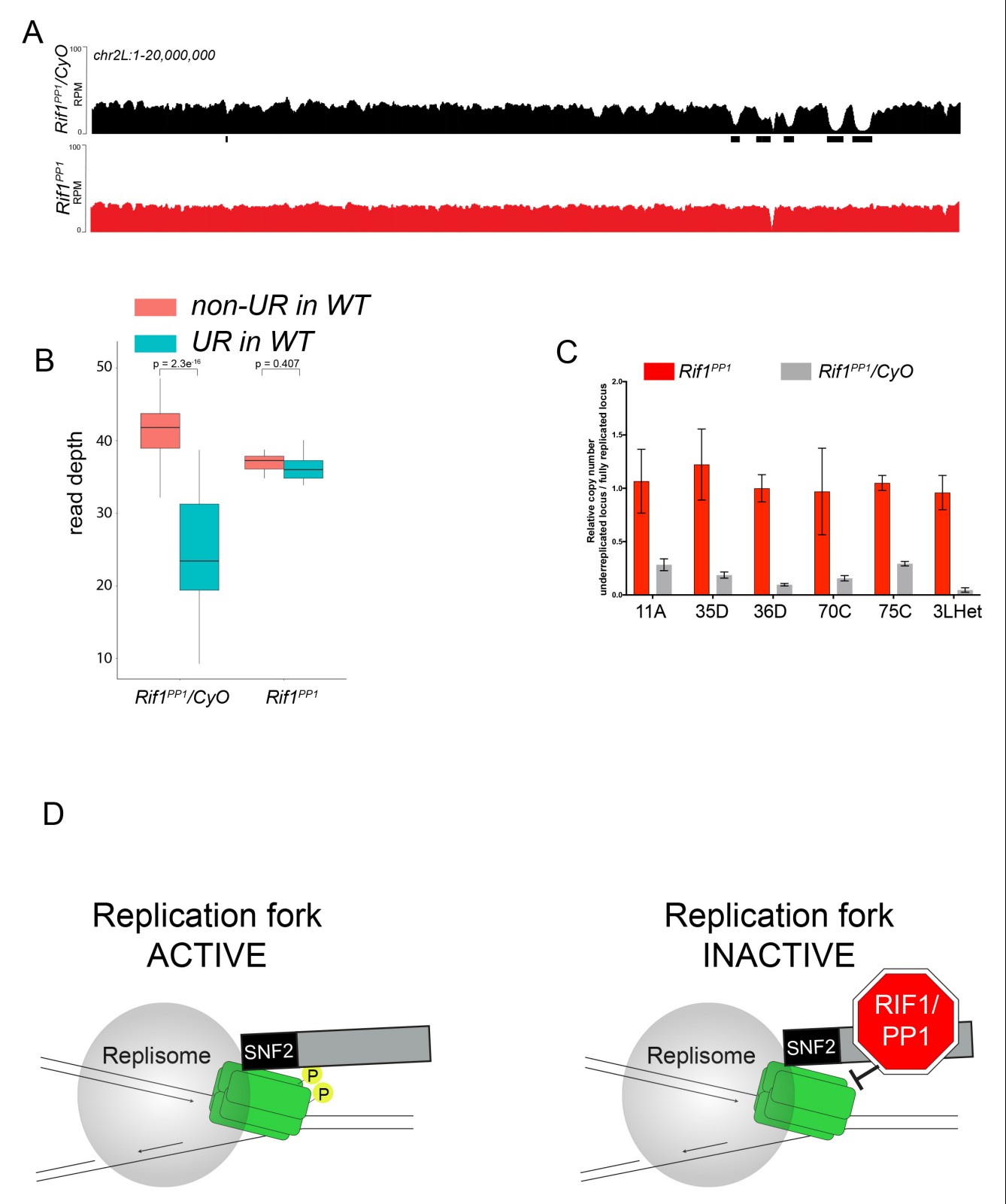

**Figure 7.** The Rif1 PP1 interaction motif is necessary to promote underreplication. (**A**) Illumina-based copy number profiles of *chr2L 1 - 20,000,000* from larval salivary glands. Black bars below each profile represent underreplicated regions identified by CNVnator. *Rif1^PP1^/CyO* was used as the wild-type control. (**B**) Average read depth in regions of euchromatic underreplication domains called in wild-type salivary glands vs. the fully replicated regions of the genome. A Welch two-sample t-test was used to determine p values. (**C**) Quantitative droplet-digital PCR (ddPCR) copy number assay for multiple

*Figure 7 continued on next page*

*Figure 7 continued*

underreplicated regions. Each bar is the average enrichment relative to a fully replicated control region for three biological replicates. Error bars are the SEM. (**D**) A new model for SUUR-mediated underreplication. In this model, SUUR serves as a scaffold to recruit a Rif1/PP1 complex to replication forks where Rif1/PP1 inhibits replication fork progression through dephosphorylation of a component of the replisome. The following source data and figure supplement are available for *Figure 7*: *Figure 7—figure supplement 1*; *Figure 7—figure supplement 2*; *Figure 7—source data 1* – raw data for 7C.
DOI: https://doi.org/10.7554/eLife.39140.030

The following source data and figure supplements are available for figure 7:

**Source data 1.** ddPCR data for *Figure 7C*.
DOI: https://doi.org/10.7554/eLife.39140.033
**Figure supplement 1.** The Rif1$^{PP1}$ protein expression level is similar to wild-type Rif1.
DOI: https://doi.org/10.7554/eLife.39140.031
**Figure supplement 2.** Genome-wide copy number profile of the *Rif1$^{PP1}$* mutant.
DOI: https://doi.org/10.7554/eLife.39140.032

remained unclear. Through biochemical, genetic, genomic and cytological approaches, we have found that SUUR recruits Rif1 to replication forks and that Rif1 is responsible for underreplication. This model is supported by several independent lines of evidence. First, SUUR associates with Rif1, and SUUR and Rif1 co-localize at sites of replication. Second, underreplication is dependent on Rif1, although *Rif1* mutants have a clear pattern of late replication in endo cycling cells. Third, SUUR localizes to replication forks and heterochromatin in a *Rif1* mutant, however, it is unable to inhibit replication fork progression in the absence of Rif1. Fourth, Rif1 controls replication fork progression and phenocopies the effect loss of SUUR function has on replication fork progression. Fifth, SUUR is required for Rif1 localization to replication forks. Critically, using the gene amplification model to separate initiation and and elongation of replication, we have shown that Rif1 can affect fork progression without altering the extent of initiation. Based on these observations, we have defined a new function of Rif1 as a regulator of replication fork progression.

## SNF2 domain and fork localization

Our work suggests that the SNF2 domain of SUUR is critical for its ability to localize to replication forks. This is based on the observation that deletion of this domain results in a protein that is unable to localize to replication forks, but still localizes to heterochromatin. SUUR has previously been shown to dynamically localize to replication forks during S phase, but constitutively binds to heterochromatin (*Kolesnikova et al., 2013*; *Nordman et al., 2014*). SUUR associates with HP1 and this interaction occurs between the central region of SUUR and HP1 (*Pindyurin et al., 2008*). Therefore, we speculate that the interaction between SUUR and HP1 is responsible for constitutive SUUR localization to heterochromatin, while a different interaction between the SNF2 domain and a yet to be defined component of the replisome, or replication fork structure itself, recruits SUUR to active replication forks during S phase.

Uncoupling of SUUR's ability to associate with replication forks and heterochromatin also provides a new level of mechanistic understanding of underreplication. Overexpression of the C-terminal two-thirds of SUUR is capable of inducing ectopic sites of underreplication. In contrast, overexpression of the SUUR's SNF2 domain, in the presence of endogenous SUUR, suppresses SUUR-mediated underreplication (*Kolesnikova et al., 2005*). Together with the data presented here, we suggest that overexpression of the SNF2 domain interferes with recruitment of full-length SUUR to replication forks, by saturating potential SUUR binding sites at the replication fork. Although the C-terminal region of SUUR is necessary to induce underreplication (*Kolesnikova et al., 2005*), the C-terminal portion of SUUR remains associated with heterochromatin in the SUUR$^{\Delta SNF}$ construct, but this protein is not sufficient to induce underreplication. We suggest that at physiological levels, the affinity of SUUR for replication forks is substantially diminished in the absence of the SNF2 domain. Our work raises questions about the biological significance of SUUR binding to heterochromatin, since without the SNF2 domain SUUR is still constitutively bound to heterochromatin, yet unable to induce underreplication. Additionally, SUUR dynamically associates with heterochromatin in mitotic cells although heterochromatin is fully replicated (*Swenson et al., 2016*).

## Rif1 controls underreplication

While trying to uncover the molecular mechanism through which SUUR is able to inhibit replication fork progression, we have uncovered an interaction between SUUR and Rif1. Through subsequent analysis, we demonstrated that Rif1 has a direct role in copy number control and that Rif1 acts downstream of SUUR in the underreplication process. Although underreplication is largely dependent on SUUR, there are several sites that display a modest degree of underreplication in the absence of SUUR (*Demakova et al., 2007*; *Sher et al., 2012*). In a *Rif1* mutant, however, these sites are fully replicated and there is no longer any detectable levels of underreplication within any regions of the genome. It is possible that Rif1 is capable of promoting underreplication through a mechanism independent of SUUR. Therefore, we conclude that Rif1 is a critical factor in driving underreplication.

Further emphasizing the critical role Rif1 plays in copy number control, we have shown that Rif1 acts downstream of SUUR in promoting underreplication. SUUR is still able to associate with chromatin in the absence of Rif1 but is unable to promote underreplication. Underreplicated regions of the genome, including heterochromatin, tend to be late replicating, raising the possibility that changes in replication timing in a *Rif1* mutant suppresses underreplication. *Rif1* mutant endo cycling cells of Drosophila display a cytological pattern of late replication, where heterochromatin is discretely replicated. While Rif1 controls replication timing in *Drosophila* and is necessary for the onset of late replication at the mid-blastula transition (*Seller and O'Farrell, 2018*), we argue that the changes in copy number associated with loss of Rif1 function are not solely due to a loss of late replication. This is supported by the clear pattern of late replication of heterochromatin in *Rif1* mutant endo cycling cells, although heterochromatin appears to be fully replicated in these cells. Previous work in mammalian polyploid cells has shown that underreplication is dependent on Rif1, which was attributed to changes in replication timing (*Hannibal and Baker, 2016*). It is important to note that Rif1-dependent changes in replication timing were not measured in this system and that many genomic regions transition from early to late replication in a *Rif1* mutant (*Foti et al., 2016*). Our work raises the possibility that Rif1 has a direct role in mammalian underreplication through a mechanism similar to that of Drosophila and may not simply be due to indirect changes in replication timing. Future work will be necessary to define the role of mammalian Rif1 in underreplication.

## Rif1 regulates replication fork progression

Our analysis of amplification loci demonstrates that Rif1 controls replication fork progression independently of initiation control, thus demonstrating that Rif1 has a specific effect on replication fork progression. Therefore, we have uncovered a new role for Rif1 in DNA metabolism as a regulator of replication fork progression and copy number control. Rif1 has been identified as part of the replisome in human cells by nascent chromatin capture, a technique that identifies proteins associated with newly synthesized chromatin (*Alabert et al., 2014*). Multiple studies have assessed whether loss of Rif1 function affects replication fork progression in yeast, mouse and human cells, but have come to different conclusions (*Alver et al., 2017*; *Cornacchia et al., 2012*; *Hiraga et al., 2017*; *Yamazaki et al., 2012*). DNA fiber assays have been used to measure fork progression in these studies and nearly all have shown that *Rif1* mutants have a slight increase in replication fork progression, although not always statistically significant. There could be several reasons for these differing results; Rif1 may control replication fork progression in specific genomic regions that may be underrepresented in some assays, Rif1 function could vary among different cell types, or sample sizes may have been too small to reach significance. Our observations, taken together with these previous studies, leave open the possibility that Rif1-mediated control of replication fork progression could be an evolutionarily conserved function of Rif1. We do not suggest that Rif1 is constitutively associated with replication forks in all cell types. Rather, Rif1 could be recruited to replication forks at a specific time in S phase, or in specific developmental contexts, to modulate the progression of replication forks and provide an additional layer of regulation of the DNA replication program.

How could SUUR and Rif1 function in concert to inhibit replication fork progression? We have shown that Rif1 retention at replication forks is dependent on SUUR. Additionally, underreplication depends on Rif1's PP1-binding motif, raising the possibility that a Rif1/PP1 complex is necessary to inhibit replication fork progression. Rif1/PP1 dephosphorylates DDK-activated helicases to control replication initiation (*Davé et al., 2014*; *Hiraga et al., 2017*; *Hiraga et al., 2014*). More recently,

however, DDK-phosphorylated MCM subunits were shown to be necessary to maintain CMG association and stability of the helicase (*Alver et al., 2017*). This result suggests that continued phosphorylation of the helicase is necessary for replication fork progression (*Alver et al., 2017*). We propose that SUUR recruits Rif1/PP1 to replication forks where it is able to dephosphorylate MCM subunits, ultimately inhibiting replication fork progression. Although this mechanism needs to be tested biochemically, it provides a framework to address the underlying molecular mechanism responsible for controlling DNA copy number and could provide new insight into the mechanism(s) Rif1 employs to regulate replication timing.

# Materials and methods

**Key resources table**

| Reagent type (species) or resource | Designation | Source or reference | Identifiers | Additional information |
|---|---|---|---|---|
| Gene (*Drosophila melanogaster*) | *Suppressor of Underreplication (SuUR)* | NA | FBgn0025355 | |
| Gene (*D. melanogaster*) | *Rap1 interacting factor 1 (Rif1)* | NA | FBgn0050085 | |
| Strain, strain background (*D. melanogaster*) | WT: *Oregon R* | | | |
| Strain, strain background (*D. melanogaster*) | *SuUR* | (*Makunin et al., 2002*) PMID: 11901119 | | $w^{118}$; *SuUR^{ES}* |
| Strain, strain background (*D. melanogaster*) | *SuUR^{ΔSNF}* | This paper | | *SuUR^{ES}*, *PBac{w^+ SuUR^{ΔSNF}}* |
| Strain, strain background (*D. melanogaster*) | *hs > FLAG-SUUR* | This paper | | *w118; hs > FLAG-SUUR* |
| Strain, strain background (*D. melanogaster*) | *hs > FLAG-SNF2* | This paper | | $w^{118}$; *hs > FLAG-SNF2* |
| Strain, strain background (*D. melanogaster*) | *Rif1^1* | This paper | | $w^{118}$; *Rif1^1* |
| Strain, strain background (*D. melanogaster*) | *Rif1^2* | This paper | | $w^{118}$; *Rif1^2* |
| Strain, strain background (*D. melanogaster*) | *Rif1^-* | This paper | | $w^{118}$; *Rif1^1/Rif1^2* |
| Strain, strain background (*D. melanogaster*) | *Rif1^{PP1}* | This paper | | $w^{118}$; *Rif1^{PP1}* |
| Cell line (*D. melanogaster*) | S2-DGRC | Drosophila Genomics Resource Center (DGRC) | embryo derived | isolate of S2 used for RNAi in the DRSC modENCODE line |

*Continued on next page*

Continued

| Reagent type (species) or resource | Designation | Source or reference | Identifiers | Additional information |
|---|---|---|---|---|
| Antibody | anti-SUUR (Guinea pig, polyclonal) | (*Nordman et al., 2011*) PMID: 25437540) | | |
| Antibody | anti-Rif1 (Guinea pig, polyclonal) | This paper | | (1:200) |
| Antibody | anti-Rif1 (Rabbit, polyclonal) | This paper | | (1:1000) |
| Antibody | HRP-anti-FLAG (Mouse, monoclonal) | Sigma-Aldrich | A8592 | (1:1000) |
| Antibody | anti-HP1 (Mouse, monoclonal) | The Developmental Studies Hybridoma Bank (DSHB) | C1A9 | (1:1000) |
| Antibody | anti-biotin (Mouse, moncolonal) | Sigma-Aldrich | SAB4200680 | (1:20,000) |
| Antibody | anti-biotin (rabbit, polyclonal) | Bethyl | A150-109A | (1:3,000) |
| Antibody | HRP-secondaries | Jackson ImmunoResearch | | (1:20,000) |
| Recombinant DNA reagent | pCaSpeR-hs | (Thummel and Pirrotta, V.)Drosophila Genomics Resource Center | | |
| Recombinant DNA reagent | pStinger | (*Barolo et al., 2000*) PMID: 11056799 | | |
| Recombinant DNA reagent | CHORI-322 (CH322-163L18) | BACPAC Resources | | |
| Recombinant DNA reagent | pET17b | Millipore-Sigma | 69663 | |
| Recombinant DNA reagent | pET17b-Rif1 (694–1094) | This paper | | Progenitors:PCR, pET17b |
| Peptide, recombinant protein | Rif1(694–1094) | This paper | | Ni-NTA purified |
| Commercial assay or kit | PLA probes | Duolink Sigma | | |
| Commercial assay or kit | PLA probemaker | Duolink Sigma | DUO92010 | |
| Commercial assay or kit | PLA Detection Reagents | Duolink Sigma | DUO92008 | |
| Chemical compound, drug | Alexa Fluor Azide 555 | Life Technologies | A20012 | |
| Chemical compound, drug | Biotin-TEG Azide | Berry and Associates | BT 1085 | |
| Chemical compound, drug | EdU (5-ethynyl-2-deoxyuridine) | Life Technologies | A10044 | |
| Software, algorithm | Sequest | Thermo Scientific | | |
| Software, algorithm | Scaffold 4.3.4 | Proteome Software | | |

*Continued*

| Reagent type (species) or resource | Designation | Source or reference | Identifiers | Additional information |
|---|---|---|---|---|
| Software, algorithm | Skyline version 4.1 | *Schilling et al. (2012)* (PMID:22454539) | | |
| Software, algorithm | deepTool 2.5.0 | *Ramírez et al. (2016)* (PMID:27079975) | | |
| Software, algorithm | CNVnator 0.3.3 | *Abyzov et al., 2011* (PMID:21324876) | | |
| Other | | | | |

## Strain list

JTN110: WT – Oregon R
JTN109: *SuUR$^-$ – w$^{118}$; SuUR$^{ES}$*
JTN038: *SuUR$^{\Delta SNF}$ – SuUR$^{ES}$, PBac{w$^+$ SuUR$^{\Delta SNF}$}*
JTN143: *w$^{118}$; hs > FLAG-SUUR*
JTN146: *w$^{118}$; hs > FLAG-SNF2*
JTN305: *w$^{118}$; Rif1$^1$*
JTN307: *w$^{118}$; Rif1$^2$Rif1$^-$ – w$^{118}$; Rif1$^1$/Rif1$^2$*
JTN292: *Rif1$^{PP1}$ – w$^{118}$; Rif1$^{PP1}$*

## BAC-mediated recombineering

BAC-mediated recombineering (*Sharan et al., 2009*) was used to delete the portion of the *SuUR* gene corresponding to the SNF2 domain. An *attB-P[acman]* clone with a 21 kb genomic region containing the *SuUR* and a *galK* insertion in the *SuUR* coding region (described in [*Nordman et al., 2014*]) was used as a starting vector. Next, a gene block (IDT) was used to replace the gal*K* cassette and generate a precise deletion within the *SuUR* gene. The resulting vector was verified by fingerprinting, PCR and sequencing. The *SuUR$^{\Delta SNF}$* BAC was injected into a strain harboring the *86* F8 landing site (Best Gene Inc.).

## Generation of heat shock-inducible, FLAG-tagged SuUR transgenic lines

The portion of the *SuUR* gene encoding the SNF2 domain (amino acids 1 to 278) was fused to the SV40 NLS (*Barolo et al., 2000*) and a 3X-FLAG tag sequence was added to the 5' end of *SuUR SNF2* sequence. The resulting construct was cloned into the pCaSpeR-hs vector, which contains a *hsp70* promoter (Thummel and Pirrotta, V.: Drosophila Genomics Resource Center), using the NotI and XbaI restriction sites. A 3X-FLAG tag sequence was added to the 5' end of of the *SuUR* coding region and cloned into the pCaSpeR-hs vector also using the NotI and XbaI restriction sites. The resulting constructs were verified by sequencing and injected into a *w$^{1118}$* strain (Best Gene Inc.).

## CRISPR mutagenesis

To generate null alleles of *Rif1*, gRNAs targeting the 5' and 3' ends of the *Rif1* gene were cloned into the pU6-BbsI plasmid as described (*Gratz et al., 2015*) using the DRSC Find CRISPRs tool (http://www.flyrnai.org/crispr2/index.html). Both gRNAs were co-injected into a *nos-Cas9* expression stock (Best Gene Inc.). Surviving adults were individually crossed to *CyO/Tft* balancer stock and *CyO*-balanced progeny were screened by PCR for a deletion of the *Rif1* locus. Stocks harboring a deletion were further characterized by sequencing. Both *Rif1$^1$* and *Rif1$^2$* mutants had substantial deletions of the *Rif1* gene and both had frame shift mutations early in the coding region. *Rif1$^1$* has a frame shift mutation at amino acid 14, whereas *Rif1$^2$* has a frame shift mutation at amino acid 11.

To generate a *Rif1* allele defective for PP1 binding, the pU6-BbsI vector expressing the gRNA targeting the 3' end of *Rif1* was co-injected with a recovery vector that contained the mutagenized SILK and RVSV (SAAK and RASA) sites with 1 kb of homology upstream and downstream of the mutagenized region. Surviving adults were crossed as above and screened by sequencing.

## Cytological analysis and microscopy

Ovaries were dissected from females fattened for two days on wet yeast in Ephrussi Beadle Ringers (EBR) medium (*Beadle and Ephrussi, 1935*). Ovaries were pulsed with 5-ethynyl-2-deoxyuridine (EdU) for 30 min, fixed in 4% formaldehyde and prepared for immunofluorescence (IF) as described (*Nordman et al., 2014*).

For IF using both anti-Rif1 and anti-SUUR antibodies, ovaries were dissected, pulsed with 50 μM EdU and fixed. Ovaries were then incubated in primary antibody (1:200) overnight at 4°C. Alexa Fluor secondary antibodies (ThermoFisher) were used at a dilution of 1:500 for 2 hr at room temperature. EdU detection was performed after incubation of the secondary antibody using Click-iT Alexa Fluor-555 or −488 (Invitrogen). All images were obtained using a Nikon Ti-E inverted microscope with a Zyla sCMOS digital camera. Images were deconvolved and processed using NIS-Elements software (Nikon).

For salivary gland IF, third instar larvae were collected prior to the wandering stage. Salivary glands were dissected in EBR, pulsed with 50 μM EdU for 30 min and fixed with 4% formaldehyde. Salivary glands were incubated in anti-HP1 antibody (Developmental Studies Hybridoma Bank; C1A9) overnight at 4°C. Alexa Fluor secondary antibodies staining and Click-iT EdU labeling were performed as described above.

## Image quantification

All images were quantified using Nikon NIS- Elements AR v4.40. To determine Rif1 and SUUR signal intensities at sites of gene amplification, Regions Of Interest (ROIs) were identified based on the EdU intensity. SUUR or Rif1 mean signal intensity was then determined within each ROI. Ten randomly selected regions outside of the nucleus were selected and the mean signal intensity for these regions were averaged to determine the background signal for each image. The average background signal was subtracted from the signal at amplified regions to normalize each image for varying amounts of background. To quantify the SUUR signal intensity at heterochromatin, SUUR ROIs were manually defined due to the their non-uniform shape. The sum intensity of the fluorescent signal within these regions were extracted. The sum signal intensity was then normalized to ROI area to account for the difference in shape of each ROI. To quantify PLA signals, ROIs were generated based on DAPI signal to mark all nuclei. PLA foci were then identified for each image and the number of foci in each DAPI ROI was determined.

## Rif1 antibody production

Rif1 antiserum was produced in guinea pigs and rabbits (Cocalico Biologicals Inc.). Briefly, a Rif1 protein fragment from residues 694 – 1094 (*Sreesankar et al., 2012*) was C-terminally six-histidine tagged and and expressed in *E. coli* Rossetta DE3 cells and purified using Ni-NTA Agarose beads (Qiagen). The purified protein was used for injection (Cocalico Biologicals Inc.) and serum was affinity purified as described (*Moore and Orr-Weaver, 1998*). Affinity purified guinea pig anti-Rif1 antibody was used for immunofluorescence.

## IP-mass spec

Flies containing heat shock-inducible *SuUR* transgenes were expanded into population cages. 0 – 24 hr embryos were collected, incubated at 37°C for 1 hr, and allowed to recover for one hour following heat shock treatment. Wild-type embryos were used as a negative control. Embryos were dechorionated in bleach and fixed for 20 min in 2% formaldehyde. Approximately 0.5 g of fixed and dechorionated embryos were used for each replicate. Embryos were disrupted by douncing in Buffer 1 (*Shao et al., 1999*), followed by centrifugation at 3000 x g for 2 min at 4°C and resuspended in lysis buffer 3 (*MacAlpine et al., 2010*). Chromatin was prepared by sonicating nuclei for a total of 40 cycles of 30' ON and 30' OFF at max power using a Bioruptor 300 (Diagnenode) with vortexing and pausing after every 10 cycles. Cleared lysates were incubated with anti-FLAG M2 affinity gel (Sigma) for 2 hr at 4°C. After extensive washing in LB3 and LB3 with 1M NaCl, proteins were eluted using 3X FLAG peptide (Sigma). Crosslinks were reversed by boiling purified material in Laemmli buffer with β-mercaptoethanol for 20 min.

Immunoprecipitated samples were separated on a 4 – 12% NuPAGE Bis-Tris gel (Invitrogen), proteins were stained with Novex colloidal Coomassie stain (Invitrogen), and destained in water.

Coomassie stained gel regions were cut from the gel and diced into 1 mm³ cubes. Proteins were reduced and alkylated, destained with 50% MeCN in 25 mM ammonium bicarbonate, and in-gel digested with trypsin (10 ng/uL) in 25 mM ammonium bicarbonate overnight at 37°C. Peptides were extracted by gel dehydration with 60% MeCN, 0.1% TFA, the extracts were dried by speed vac centrifugation, and reconstituted in 0.1% formic acid. Peptides were analyzed by LC-coupled tandem mass spectrometry (LC-MS/MS). An analytical column was packed with 20 cm of C18 reverse phase material (Jupiter, 3 µm beads, 300 Å, Phenomenox) directly into a laser-pulled emitter tip. Peptides were loaded on the capillary reverse phase analytical column (360 µm O.D. x 100 µm I.D.) using a Dionex Ultimate 3000 nanoLC and autosampler. The mobile phase solvents consisted of 0.1% formic acid, 99.9% water (solvent A) and 0.1% formic acid, 99.9% acetonitrile (solvent B). Peptides were gradient-eluted at a flow rate of 350 nL/min, using a 120 min gradient. The gradient consisted of the following: 1 – 3 min, 2% B (sample loading from autosampler); 3 – 98 min, 2 – 45% B; 98 – 105 min, 45 – 90% B; 105 – 107 min, 90% B; 107 – 110 min, 90–2% B; 110 – 120 min (column re-equilibration), 2% B. A Q Exactive HF mass spectrometer (Thermo Scientific), equipped with a nanoelectrospray ionization source, was used to mass analyze the eluting peptides using a data-dependent method. The instrument method consisted of MS1 using an MS AGC target value of 3e6, followed by up to 15 MS/MS scans of the most abundant ions detected in the preceding MS scan. A maximum MS/MS ion time of 40 ms was used with a MS2 AGC target of 1e5. Dynamic exclusion was set to 20 s, HCD collision energy was set to 27 nce, and peptide match and isotope exclusion were enabled. For identification of peptides, tandem mass spectra were searched with Sequest (Thermo Fisher Scientific) against a *Drosophila melanogaster* database created from the UniprotKB protein database (www.uniprot.org). Search results were assembled using Scaffold 4.3.4 (Proteome Software).

## Genome-wide copy number profiling

Embryos were collected immediately after 2 hr of egg laying. Salivary glands were dissected in EBR from 50 wandering 3rd instar larvae per genotype and flash frozen. Ovaries were dissected from females fattened for 2 days on wet yeast in EBR and 50 stage 13 egg chambers were isolated for each genotype and flash frozen. Tissues were thawed on ice, resuspended in LB3 and dounced using a Kontes B-type pestle. Dounced homogenates were sonicated using a Bioruptor 300 (Diagenode) for 10 cycles of 30′ on and 30′′ off at maximal power. Lysates were treated with RNase and Proteinase K and genomic DNA was isolated by phenol-chloroform extraction. Illumina libraries were prepared using NEB DNA Ultra II (New England Biolabs) following the manufacturers protocol. Barcoded libraries were sequenced using Illumina NextSeq500 platform.

## Bioinformatics

Reads were mapped to the Drosophila genome (BDGP Release 6) using BWA-MEM with default parameters (*Li and Durbin, 2009*). CNVnator 0.3.3 was used for the detection of underreplicated regions using a bin size of 1000 (*Abyzov et al., 2011*). Regions were identified as underreplicated if they were not identified as underreplicated in 0 – 2 hr embryonic DNA and were greater than 10 kb in length. The number of reads for underreplicated regions was called by using bedtools multicov tool for the underreplicated and uncalled regions. Average read depth per region was determined by multiplying the number of reads in a region by the read length and dividing by the total region length. Read depth was normalized between samples by scaling the total reads obtained per sample. Statistical comparison between the regions was with a t-test. For read depth in pericentric heterochromatin regions, the chromatin arm was binned into 10 kb windows and the number of reads for each window was called using bedtools multicov using only uniquely mapped reads.

Half maximum analysis of amplicon copy number profiles was performed as described previously (*Alexander et al., 2015*; *Nordman et al., 2014*). Briefly, $log_2$ ratios were generated using bamCompare from deepTools 2.5.0 (Ramírez et al., n.d.) by comparing stage 13 follicle cell profiles to a 0 – 2 hr embryo sample. Smoothed $log_2$-transformed data was used to determine the point of maximum copy number associated with each amplicon. The chromosome coordinate corresponding to half the maximum value for each arm of the amplicon was then determined.

## Copy number analysis by droplet-digital PCR (ddPCR)

Genomic DNA was extracted from salivary glands isolated from wandering third instar larvae as described above. Primer sets annealing to the mid-point of the indicated UR regions were used (previously described in [*Nordman et al., 2014*; *Sher et al., 2012*]). ddPCR was performed according to manufacture's recommendations (BioRad). All ddPCR reactions were performed in triplicate from three independent biological replicates. The concentration value for each set of primers in an under-replicated domain was divided by the concentration value of a fully replicated control to generate the bar graph. Error bars represent the SEM.

## Western blotting

Ovaries were dissected from females fattened for 2 days on wet yeast and suspended in Laemmli buffer supplemented with DTT. Ovaries were homogenized and boiled and extracts were loaded on a 4–20% Mini-PROTEAN TGX Stain-Free gel (BioRad). After electrophoresis the gel was activated and imaged according to the manufacturers recommendations. Protein was transferred to a PDVF membrane using a Trans-Blot Turbo Transfer System (BioRad). After blocking and incubation with antibodies, blots were imaged using an Amersham 600 CCD imager.

## iPOND mass spectrometry

We obtained *D. melanogaster* S2 cells directly from the Drosophila Genomics Resource Center (DGRC). Cells were checked for mycoplasma contamination by PCR. S2 cells propagated as recommended by the DGRC. *Drosophila* S2 cells were grown in Schneider's Drosophila Medium with 10% heat-inactivated FBS (Gemini Bio Products) and 100 units/mL Penicillin-Streptomycin (Life Technologies). For each biological replicate, $5 \times 10^8$ cells were pulsed with 10 μM EdU and immediately fixed in 2% formaldehyde (pulse samples) or pulsed with 10 μM EdU for 10 minutes and chased with 100 μM Thymidine for 30 min prior to fixation (chase samples). iPOND purifications were done according to (*Dungrawala and Cortez, 2014*) with the exception that LB3 was used in place of RIPA buffer. Purifications were processed for mass spectrometry as described above.

To quantify protein abundance in by mass spectrometry, raw mass spectrometry data were imported into Skyline version 4.1 (*Schilling et al., 2012*). Chromatographic traces were manually inspected for proper peak picking and where necessary adjusted manually in the chromatographic window. Only matching isotopic envelopes that had an error <5 ppm, an isotope dot product >0.9, and similar retention times between samples were used. MS1 peak areas from each peptide and observed charge state were summed to get the intensity for a given protein. To account for variation between samples, each sample was normalized to histone H3 summed areas.

## Proximity Ligation Assay (PLA) with nascent DNA

S2 cells were grown in Schneider's Drosophila Medium with 10% heat-inactivated FBS (Gemini Bio Products) and 100 units/mL Penicillin Streptomycin (Life Technologies). For nascent DNA PLA, asynchronously growing S2 cells were seeded onto Concanavalin A-coated coverslips. After attaching to coverslips for 1 hr, cells were pulsed with 125 μM EdU for 10 min. Cells were washed with PBS and fixed in 4% paraformaldehyde for 15 min, then permeabilized in PBS + 0.25% Triton X-100 for 60 min. The cells were biotinylated using standard click chemistry conditions for 30 min. After washing 3 times, blocking was performed for 1 hr with Duolink blocking solution. Cells were incubated with their primary antibody overnight at 4°C. The following day, cells were washed in Duolink Wash Buffer A, then incubated at 37°C with the appropriate Plus and Minus PLA probes at a 1:5 dilution. After an hour, the cells were washed in Wash Buffer A twice, ligation buffer was made at 1:40 dilution and incubated for 30 min at 37°C. Cells were washed 2x in Wash Buffer A, then incubated in amplification buffer at 1:80 for 100 min at 37°C. Slides were washed in Wash Buffer B, then 1:100 dilution of Wash Buffer B before being mounted in Duolink In Situ Mounting Media with DAPI.

## Data access

Data sets described in this manuscript can be found under the GEO accession number: GSE114370.

## Acknowledgements

We thank Kristie Rose and Hayes McDonald at the Vanderbilt Proteomics core for mass spectrometry and Olivia Koues from the VANTAGE core at Vanderbilt for Illumina sequencing. We would like to thank Terry Orr-Weaver, Stephen Bell, Katherine Friedman, James Dewar, Dave Cortez and members of the Nordman lab for providing critical comments on the manuscript. We thank Brooke Hamilton for assistance in generating the *Rif1* mutants. This work was supported by an NIH award P30 AI110527 to SM and an NIH R00 award 5R00GM104151 to JTN.

## Additional information

### Funding

| Funder | Grant reference number | Author |
|---|---|---|
| National Institutes of Health | 5R00GM104151 | Jared T Nordman |
| National Institutes of Health | P30 AI110527 | Simon Mallal |

The funders had no role in study design, data collection and interpretation, or the decision to submit the work for publication.

### Author contributions

Alexander Munden, Zhan Rong, Formal analysis, Investigation, Visualization; Amanda Sun, Formal analysis, Investigation, Methodology; Rama Gangula, Formal analysis, Methodology; Simon Mallal, Supervision, Methodology; Jared T Nordman, Conceptualization, Formal analysis, Supervision, Funding acquisition, Validation, Investigation, Visualization, Writing—original draft

### Author ORCIDs

Jared T Nordman iD http://orcid.org/0000-0002-6612-3201

### Decision letter and Author response

Decision letter https://doi.org/10.7554/eLife.39140.040
Author response https://doi.org/10.7554/eLife.39140.041

## Additional files

### Supplementary files

• Supplementary file 1. Underreplicated regions called by CNVnator.
DOI: https://doi.org/10.7554/eLife.39140.034

• Supplementary file 2. Half-max copy number analysis.
DOI: https://doi.org/10.7554/eLife.39140.035

• Transparent reporting form
DOI: https://doi.org/10.7554/eLife.39140.036

### Data availability

Sequencing data have been deposited in GEO under accession code GSE114370

The following dataset was generated:

| Author(s) | Year | Dataset title | Dataset URL | Database, license, and accessibility information |
|---|---|---|---|---|
| Nordman JT, Munden A | 2018 | Rif1 inhibits replication fork progression and controls copy number in Drosophila. | https://www.ncbi.nlm.nih.gov/geo/query/acc.cgi?acc=GSE114370 | Publicly available at the NCBI Gene Expression Omnibus (accession no: GSE114370). |

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
