## [Decision Letter]

Thank you for submitting your article "Rif1 inhibits replication fork progression and controls DNA copy number in *Drosophila*" for consideration by *eLife*. Your article has been reviewed by two peer reviewers, and the evaluation has been overseen by a Reviewing Editor and Jessica Tyler as the Senior Editor. The reviewers have opted to remain anonymous.

The reviewers have discussed the reviews with one another and the Reviewing Editor has drafted this decision to help you prepare a revised submission. Both reviewers found the work of general interest but each felt that certain biochemical experiments needed to be done to shore up the conclusions drawn from the genetics.

This manuscript reports the interesting finding that SuUR and Rif1 have a relationship to regulate DNA replication in different cell types of *Drosophila*. The manuscript is well written, the experiments well controlled, and the data well documented. It also reports new transgenes/mutants for both SuUR and Rif1. The data is convincing that SuUR and Rif1 co-purify, with certain caveats as to if the interactions are as direct as stated from embryos, and that both proteins promote under-replication in salivary glands and restrict the developmental window for fork progression during amplification. It is also convincing that Rif1 co-localizes to amplicon forks by immunnofluorescent labeling, that SuUR localization to these forks does not depend on Rif1, and that mutation of a putative PP1 interacting domain in Rif1 also results in more heterochromatic replication. The authors propose that the C terminal two-thirds of SUUR is important for interacting with an effector protein that suppresses replication fork progression. Thus they identify the protein as Rif1 and demonstrate that loss of Rif1 produces a phenotype similar to loss of SUUR: reversal of under-replication in endocyling cells. They show that Rif1 associates with elongating replication forks, that the association is dependent on SUUR and that the PP1 interacting motif of Rif1 is important for the suppression of replication fork progression.

As stated above, what is less clear is whether the SuUR and Rif1 interaction is direct, whether Rif1 inhibition of amplicon forks is "direct", and the extent to which fork progression versus origin activity explain the effect of Rif1 mutants on under-replication in polyploid salivary gland. These rather opposing ideas should be discussed. Indeed, the data presented suggest that Rif1 localization to forks and its effect on replication is only partially dependent on SuUR. In general, there is a deficit of biochemical approaches (e.g. iPOND, DNA fiber analysis, pull down) to bolster the conclusions from the genetic and cytological data. A previous report for a Rif1 effect on forks in frog extracts also partially compromises novelty. Therefore, although the manuscript reports some interesting findings, there are a number of changes that would address rigor. Major points for revision are put into context below and will be required for revisions.

Essential revisions:

1) The authors conclude that Rif1 affects "fork progression." This is based on two observations: a) Rif1 mutant salivary glands still have a late replication pattern of EdU incorporation but increased copy number of some heterochromatic DNA, b) IF co-localization of Rif1 with forks and an extended developmental time window of replication fork migration at amplicons in Rif1 mutants. The first argument in salivary glands is very indirect. Only a subset of the under-replicated loci in salivary glands are affected, others remain fully under-replicated, and it is not surprising that some of the late replication program would be retained. Moreover, it is not directly addressed to what extent Rif1's known activity at origins explains the increased DNA copy number in the mutant, e.g. activation of dormant origins. The second observation for amplification shows that the developmental time window is extended but does not address whether fork rate is affected. It seems like the manuscript stops short of doing the key experiments to show Rif1 directly affects fork rate/stability, i.e. iPOND and DNA fiber analysis during amplification and genomic replication. One or another of such experimental approaches needs to be done.

2) The authors conclude that SNF2 domain is essential for SUUR function and replication fork localization. a) The authors should state in the Results that they are over-expressing the mutant using the hsp70 promoter. b) Does deletion of the SNF2 domain affect SUUR protein abundance (despite over-expression)? In Figure 1E, It appears some localization to heterochromatin remains but looks reduced. Is it possible to do a Western or other quantification of protein abundance? Going beyond those controls, has it been addressed whether the FLAG-SNF2 is sufficient for fork localization?

3) The SUUR^ΔSNF^ localizes to heterochromatin but not the amplifying loci. Is the protein as stable and expressed at the same level as wild type SUUR?

4) The SNF2 domain of SUUR did not co-IP Rif1. This is an important negative control. However, the co-IP was done after formaldehyde cross-linking like in a ChIP experiment. Are the authors sure that the SNF2 domain of SUUR is expressed to the same level as full length SUUR and localizes properly to the chromatin? In the same vein, is it possible that the interaction between SUUR and Rif1 will be lost if the samples were DNAsed? If the latter, then the idea that SUUR directly recruits Rif1 may be incorrect. SUUR may set up conditions at the fork that enable Rif1 to be recruited.

5) A notable omission is that the authors do not mention whether Rif1 mutants have a DNA replication phenotype in diploid cells. Perhaps a comment on this even if not known would be worth mentioning

6) The Rif1-PP1 mutant is believed to not interact with PP1 based on evolutionary arguments. A simple demonstration that WT Rif1 co-IPs PP1 (the authors have a very nice antibody to Rif1) and that the Rif1-PP1 mutant does not do so will greatly strengthen the hypothesis. Even with that result, it is still possible that the effector downstream of Rif1 is another protein (not PP1) that interacts with Rif1 near the same site as PP1. It is best to leave that possibility open.

---

## [Author Response]

Essential revisions:1) The authors conclude that Rif1 affects "fork progression." This is based on two observations: a) Rif1 mutant salivary glands still have a late replication pattern of EdU incorporation but increased copy number of some heterochromatic DNA, b) IF co-localization of Rif1 with forks and an extended developmental time window of replication fork migration at amplicons in Rif1 mutants. The first argument in salivary glands is very indirect. Only a subset of the under-replicated loci in salivary glands are affected, others remain fully under-replicated, and it is not surprising that some of the late replication program would be retained. Moreover, it is not directly addressed to what extent Rif1's known activity at origins explains the increased DNA copy number in the mutant, e.g. activation of dormant origins. The second observation for amplification shows that the developmental time window is extended but does not address whether fork rate is affected. It seems like the manuscript stops short of doing the key experiments to show Rif1 directly affects fork rate/stability, i.e. iPOND and DNA fiber analysis during amplification and genomic replication. One or another of such experimental approaches needs to be done.

Our conclusion that Rif1 affects replication fork progression is solely based on the observation that *Rif1* mutants exhibit increased replication fork progression in amplifying follicle cells without altering initiation of DNA replication. The observation that loss of Rif1 function suppresses under-replication merely raised the possibility that Rif1 could control replication fork progression in endo cycling cells based on what is known about SUUR function.

We would like to make clear that loss of Rif1 function results in increased fork progression at all amplicons without altering the developmental time window of gene amplification. We have measured the distribution of egg chambers in wild-type and Rif1 mutant follicle cells to make this clearer. If the developmental window for gene amplification was increased, we would expect to see an increase in amplification stage egg chambers in the Rif1 mutant. In contrast, we see no difference in egg chamber between wild-type and the Rif1 mutant. These data have been added as a Figure 4—figure supplement 1. The increased percentage of EdU+ stage 13 follicle cells we observe in the Rif1 mutant reflects that forks are more stable and therefore extend the window of EdU incorporation. Importantly, this prolonged EdU incorporation is occurring within the same 7.5 hour time window of gene amplification as in wild-type follicle cells. We have also changed the text to distinguish between the period of EdU incorporation and the developmental window of gene amplification to make this crucial point more clear.

We agree with the reviewers that DNA fiber analysis would be a convenient way to compare directly fork rates in under-replicated regions of wild-type and Rif1 mutant endo cycling cells. Under-replicated domains are where we predict Rif1 will have the strongest effect. Unfortunately, we would not be able to interpret this experiments because 1) UR domains are largely blocked for replication in wild-type endo cycling cells and 2) while UR is not complete, it is not known if the molecules that escape UR are at all affected by SUUR/Rif1. Therefore, we lack the proper, and interpretable, control for track length in UR domains for wild-type endo cycling cells. For example, the only visible tracks that we ultimately score within under-replicated domains may be those that are not affected by SUUR/Rif1.

We did, however, perform iPOND as the reviewers suggested. It is not feasible to perform iPOND in amplifying follicle cells due the large amount of hand-dissected material required for biochemical analysis. Therefore, we performed iPOND mass-spec in *Drosophila* cultured S2 cells and quantified Rif1 protein abundance in pulse and chase samples along with PCNA and Histone H1 controls, which should be enriched in the pulse and chase samples respectively. Our results reveal that Rif1 is enriched at replication forks in S2 cells and has been included as Figure 6—figure supplement 2A. Additionally, we used a proximity ligation assay (PLA)-based technique to monitor Rif1 localization at newly synthesized DNA. This recently developed technique, is the cytological equivalent of iPOND where the proximity of a protein of interest is monitored relative to newly synthesized DNA. Here show that Rif1 is associated with newly replicated DNA is cultured cells (Figure 6—figure supplement 2B).

2) The authors conclude that SNF2 domain is essential for SUUR function and replication fork localization. a) The authors should state in the Results that they are over-expressing the mutant using the hsp70 promoter. b) Does deletion of the SNF2 domain affect SUUR protein abundance (despite over-expression)? In Figure 1E, It appears some localization to heterochromatin remains but looks reduced. Is it possible to do a Western or other quantification of protein abundance? Going beyond those controls, has it been addressed whether the FLAG-SNF2 is sufficient for fork localization?

Our observation that the SNF2 domain is essential for SUUR function was based on our analysis of the *SUUR^ΔSNF^* mutant, where we deleted the SNF2 domain of SUUR and expressed this mutant form of *SuUR* under control of its own promoter. This was described in the first paragraph of the Results subsection “The SNF2 domain is essential for SUUR function and replication fork localization”. We did, however, use the *hsp70* promoter to overexpress the SNF2 domain and full length SUUR protein for IP-mass spec experiments where we identified an interaction between SUUR and Rif1. We have added the following sentence to make this clear in the Results section “We generated flies that expressed FLAG-tagged full length SUUR or the SNF2 domain of SUUR under control of the hsp70 promoter, immunoprecipitated these constructs and identified associated proteins through mass spectrometry.” We also added included the following statement in the Materials and methods section “The resulting construct was cloned into the pCaSpeR-hs vector, which contains a *hsp70* promoter (Thummel and Pirrotta, V.: *Drosophila* Genomics Resource Center”

Unfortunately, no reagents exist to monitor endogenous SUUR by Western blotting, which limits our ability to ensure that the SUUR^ΔSNF^ mutant is maintained at the same levels as wild-type SUUR. What we were able to do, however, is quantify the signal intensity of SUUR and SUUR^ΔSNF^ proteins at heterochromatin by quantitative IF. This analysis revealed no significant difference in levels of SUUR and SUUR^ΔSNF^ protein signal at heterochromatin, while there is a dramatic difference in SUUR and SUUR^ΔSNF^ protein levels at sites of amplification. This data has now been added as Figure 1—figure supplement 2.

We currently do not know if FLAG-SNF2 is sufficient for fork localization. When we monitor FLAG-SNF2 localization in follicle cells, we see a strong staining throughout the nucleus. It is possible that this construct is at replication forks, but the overall chromatin signal may be masking signal at amplification regions.

3) The SUUR^ΔSNF^ localizes to heterochromatin but not the amplifying loci. Is the protein as stable and expressed at the same level as wild type SUUR?

See above. There are no available reagents to monitor endogenous SUUR protein by Western blotting. We have performed quantitative IF and shown that there is no difference in the protein signal of SUUR and SUUR^ΔSNF^ at heterochromatin.

4) The SNF2 domain of SUUR did not co-IP Rif1. This is an important negative control. However, the co-IP was done after formaldehyde cross-linking like in a ChIP experiment. Are the authors sure that the SNF2 domain of SUUR is expressed to the same level as full length SUUR and localizes properly to the chromatin? In the same vein, is it possible that the interaction between SUUR and Rif1 will be lost if the samples were DNAsed? If the latter, then the idea that SUUR directly recruits Rif1 may be incorrect. SUUR may set up conditions at the fork that enable Rif1 to be recruited.

We agree with the reviewers that we have not provided any evidence that the interaction between Rif1 and SUUR is direct. We initially described the interaction as a ‘physical interaction’ to distinguish it from a genetic interaction. We have now repeated the IP in the absence of crosslinking, +/- benzonase, to support the idea that SUUR recruits Rif1 to replication forks rather than setting up a condition at the fork that enables Rif1 to be recruited. Our new results show a robust interaction between SUUR and Rif1 using NP40 extraction and treating extracts with benzonase to cleave DNA. We’ve combined these data with our initial Table 1 as a single figure (Figure 2). We have not been able to purify full-length SUUR for in vitro binding assays to test directness, however, our new results provide much more compelling evidence that SUUR and Rif1 are closely associated. We have described SUUR and Rif1 as being in the same complex rather than a physical interaction to avoid confusion.

5) A notable omission is that the authors do not mention whether Rif1 mutants have a DNA replication phenotype in diploid cells. Perhaps a comment on this even if not known would be worth mentioning

While preparing this manuscript, Seller and O’Farrell published an elegant paper on the role of Rif1 in the diploid cells of the early embryo (Seller and O’Farrell, 2018). They demonstrate that Rif1 localizes to heterochromatin and is necessary to establish a program of late replication at the MBT. We have cited this manuscript and commented when appropriate.

6) The Rif1-PP1 mutant is believed to not interact with PP1 based on evolutionary arguments. A simple demonstration that WT Rif1 co-IPs PP1 (the authors have a very nice antibody to Rif1) and that the Rif1-PP1 mutant does not do so will greatly strengthen the hypothesis. Even with that result, it is still possible that the effector downstream of Rif1 is another protein (not PP1) that interacts with Rif1 near the same site as PP1. It is best to leave that possibility open.

In the Seller and O’Farrell manuscript referenced above, it was shown that Rif1 can co-IP PP1. Furthermore, Sreesankar et al., previously demonstrated that an HA tagged version of PP1 could co-IP Rif1 (Sreesankar et al., 2015). We have tried to co-IP PP1 with Rif1 using our antibody and have found, unfortunately, that our antibody blocks that interaction between Rif1 and PP1. The epitope we used to generate the antibodies included the PP1-binding motif. Regardless, we agree with the reviewers that we cannot exclude the possibility that another downstream effector molecule other then PP1 interacts with this region. We have raised this possibility in the Results and Discussion sections and have been careful to say that the PP1-binding motif is essential for Rif1 function rather than PP1 binding or an interaction between Rif1 and PP1.